# CDK12 loss in cancer cells affects DNA damage response genes through premature cleavage and polyadenylation

Malgorzata Krajewska[1,2,11], Ruben Dries[1,2,3,11], Andrew V. Grassetti[4], Sofia Dust[5], Yang Gao[1,2], Hao Huang [1,2], Bandana Sharma[1], Daniel S. Day[6], Nicholas Kwiatkowski[7], Monica Pomaville[1], Oliver Dodd[1], Edmond Chipumuro[1], Tinghu Zhang[7], Arno L. Greenleaf[8], Guo-Cheng Yuan [3,9], Nathanael S. Gray [7,10], Richard A. Young [6], Matthias Geyer[5], Scott A. Gerber[4] & Rani E. George[1,2]

Cyclin-dependent kinase 12 (CDK12) modulates transcription elongation by phosphorylating the carboxy-terminal domain of RNA polymerase II and selectively affects the expression of genes involved in the DNA damage response (DDR) and mRNA processing. Yet, the mechanisms underlying such selectivity remain unclear. Here we show that CDK12 inhibition in cancer cells lacking CDK12 mutations results in gene length-dependent elongation defects, inducing premature cleavage and polyadenylation (PCPA) and loss of expression of long (>45 kb) genes, a substantial proportion of which participate in the DDR. This early termination phenotype correlates with an increased number of intronic polyadenylation sites, a feature especially prominent among DDR genes. Phosphoproteomic analysis indicated that CDK12 directly phosphorylates pre-mRNA processing factors, including those regulating PCPA. These results support a model in which DDR genes are uniquely susceptible to CDK12 inhibition primarily due to their relatively longer lengths and lower ratios of U1 snRNP binding to intronic polyadenylation sites.

[1] Department of Pediatric Hematology/Oncology, Dana-Farber Cancer Institute and Boston Children's Hospital, Boston, MA 02115, USA. [2] Department of Pediatrics, Harvard Medical School, Boston, MA 02115, USA. [3] Departments of Biostatistics and Computational Biology, Dana-Farber Cancer Institute, Boston, MA 02215, USA. [4] Department of Molecular and Systems Biology, Geisel School of Medicine at Dartmouth, Lebanon, NH 03756, USA. [5] Institute of Structural Biology, University of Bonn, 53127 Bonn, Germany. [6] Whitehead Institute for Biomedical Research, Massachusetts Institute of Technology, Cambridge, MA 02142, USA. [7] Department of Cancer Biology, Dana-Farber Cancer Institute, Boston, MA 02215, USA. [8] Department of Biochemistry, Duke University Medical Center, Durham, NC 27710, USA. [9] Harvard School of Public Health, Boston, MA 02115, USA. [10] Department of Biological Chemistry and Molecular Pharmacology, Harvard Medical School, Boston, MA 02115, USA. [11]These authors contributed equally: Malgorzata Krajewska, Ruben Dries. Correspondence and requests for materials should be addressed to R.E.G. (email: Rani_George@dfci.harvard.edu)

Eukaryotic gene transcription is facilitated by the orchestrated action of transcriptional cyclin-dependent kinases (CDKs) and associated pre-mRNA processing factors[1,2]. Transcriptional CDKs phosphorylate the carboxy-terminal domain (CTD) of RNA Polymerase II (Pol II) which serves as a platform for the recruitment of factors controlling transcriptional and post-transcriptional events. During transcription initiation, CDK7, a subunit of TFIIH, phosphorylates serine 5 of the CTD[3]; subsequently, the release of paused Pol II and the transition to elongation is mediated by CDK9, a subunit of pTEFb, which phosphorylates the CTD at serine 2[4]. Studies in yeast and metazoans have shown that another transcriptional kinase, CDK12, together with its associating partner, cyclin K, modifies serine 2 of the Pol II CTD[5–7]. A second, less-studied metazoan ortholog of yeast Ctk1 in human cells is CDK13, which shares a largely conserved kinase domain with CDK12[6]. Although the biological role of CDK13 is not known, its sequence similarity with CDK12 predicts some degree of overlap between these kinases. In contrast to other transcriptional CDKs, both CDK12 and CDK13 contain additional arginine/serine-rich (RS) domains that are critical for proteins involved in processing premature RNA[8,9]. However, based on genetic depletion studies, CDK12 but not CDK13 has been reported to control the expression of DNA damage response (DDR) genes[6,10]. The selective regulation of these genes by CDK12 is also evident in cancers with loss-of-function CDK12 mutations, such as high-grade serous ovarian carcinoma and metastatic castration-resistant prostate cancer, where a BRCAness phenotype with genomic instability sensitizes cells to DNA cross-linking agents and poly (ADP-ribose) polymerase (PARP) inhibitors[11–13]. Similarly, suppression of wild-type CDK12 in Ewing sarcoma cells driven by the EWS/FLI fusion oncoprotein using THZ531[14] (a selective inhibitor of CDK12/13) also led to the decreased expression of DDR genes[15]. Hence, CDK12 loss of function, whether spontaneous or induced, appears to preferentially affect genes that have prominent roles in DNA repair.

Despite growing knowledge of CDK12 function in cancer cells and the availability of selective CDK12/13 inhibitors, the molecular basis for the selective effects of this kinase on DDR genes remains unclear. This deficit could have important implications for understanding distinctions among transcriptional CDKs and devising treatments for cancers that rely on aberrant transcription and/or genomic instability for their sustained survival and growth. Thus, using MYCN-amplified neuroblastoma (NB) as a solid tumor model characterized by constitutive transcriptional upregulation[16], and genomic instability[17,18], but lacking CDK12 mutation[19], we demonstrate a mechanistic link between the structural properties of DDR genes and their susceptibility to CDK12 inhibition.

## Results

**CDK12/13 inhibition with THZ531 is cytotoxic to NB cells.** To understand the preferential effect of CDK12 on the DDR, we first determined whether we could abrogate its activity by using THZ531. This covalent inhibitor binds to unique cysteine residues outside the canonical kinase domains of both CDK12 and 13 (Cys1039 and Cys1017, respectively), resulting in their prolonged and irreversible inactivation[14]. Importantly, no other transcriptional CDK, including CDK9, contains a cysteine at a similar position and hence is not targeted by this inhibitor[14].

We observed strong selectivity and cytotoxicity in NB cells compared to nontransformed cells (Fig. 1a, Supplementary Fig. 1a). Decreased sensitivity was also observed in Kelly E9R NB cells expressing a point mutation at the CDK12 Cys1039 THZ531 binding site[20] [IC$_{50}$ = 400 nM compared to 60 nM in Kelly wild-type (WT) cells], suggesting that inhibition of CDK13 alone did not affect cell viability. Target engagement studies using a biotinylated derivative of the compound (bio-THZ531) revealed consistently decreased binding to CDK12 and CDK13 after treatment with THZ531, indicating that these kinases are indeed targets of this inhibitor (Supplementary Fig. 1b). THZ531 treatment led to apoptosis as well as G2/M cell cycle arrest in these cells (Supplementary Fig. 1c–e). The sensitivity to THZ531 extended to both MYCN-amplified and nonamplified NB cells; in the latter, the addition of an ABCB1 drug efflux pump inhibitor (tariquidar) was necessary to overcome high expression of this protein and subsequent inhibitor efflux[20,21] (Supplementary Fig. 1f). Despite the role of CDK12 in transcription elongation[5], THZ531 induced variable dose- and time-dependent decreases in Pol II Ser2 phosphorylation, with minimal effects on Ser5 or 7 phosphorylation (Fig. 1b). However, we observed striking downregulation of termination-associated Pol II threonine (Thr4)[22,23] phosphorylation, indicating that distal elongation was affected (Fig. 1b). Together, these results indicate that THZ531, by binding to CDK12/13, induces cytotoxicity in NB cells through effects on transcription elongation.

**CDK12 inhibition preferentially affects DDR genes.** CDK12 inhibition has been shown to affect the expression of genes involved in the DDR[6,15]. To determine whether similar effects are produced by our selective inhibitor in NB cells, we analyzed the gene expression profiles of cells treated with and without THZ531 for 6 h, a time point at which there were little or no confounding effects due to cell cycle changes (Supplementary Fig. 1e). Unlike the effects seen with THZ1[16], predominantly an inhibitor of CDK7 with some activity against CDK12/13[24], we failed to observe a complete and global transcriptional shutdown in THZ531-treated NB cells; instead, only 57.4% of the transcripts were downregulated (n = 10,707), with 0.35% (n = 66) upregulated [false discovery rate (FDR) < 0.05] (Supplementary Fig. 2a; Supplementary Data 1). Consistent with earlier studies[14,15], THZ531 led to significant downregulation of both transcription-associated and DDR genes (Fig. 1c, Supplementary Fig. 2b, c), the latter of which were primarily associated with homologous recombination (HR) repair and are crucial for the maintenance of genome stability, including BRCA1, BARD1 and RAD51[25] (Fig. 1c, Supplementary Fig. 2c, d). To determine whether these effects were due to inhibition of CDK12 or 13, we depleted the expression of each kinase individually in NB cells, and in keeping with prior studies[6,10], observed selective downregulation of DDR genes with CDK12 but not CDK13 knockdown (KD) (Supplementary Fig. 2e). Additionally, the expression of DDR genes was not significantly affected in Kelly E9R THZ531-resistant cells, further implicating the selective role of CDK12 in regulating the DDR (Fig. 1d). Consistent with these observations, THZ531 also led to increased DNA damage with elevated γ-H2AX levels (Fig. 1e, Supplementary Fig. 2f) and decreased radiation-induced RAD51 foci, indicating defects in DNA repair (Fig. 1f). Thus, our findings indicate that DDR genes are selectively affected by THZ531 and that such regulation is driven predominantly by CDK12.

**CDK12/13 inhibition leads to an elongation defect.** The transcriptional effects of CDK12/13 inhibition with THZ531, including downregulation of the steady-state expression of DDR genes, occurred as early as 6 h. post-treatment and independently of cell cycle changes (Supplementary Fig. 1e). This result, plus the fact that CDK12 and 13 have been implicated in pre-mRNA processing[10,26,27] where observable changes are likely to occur within minutes or hours, indicated that further analysis of steady-

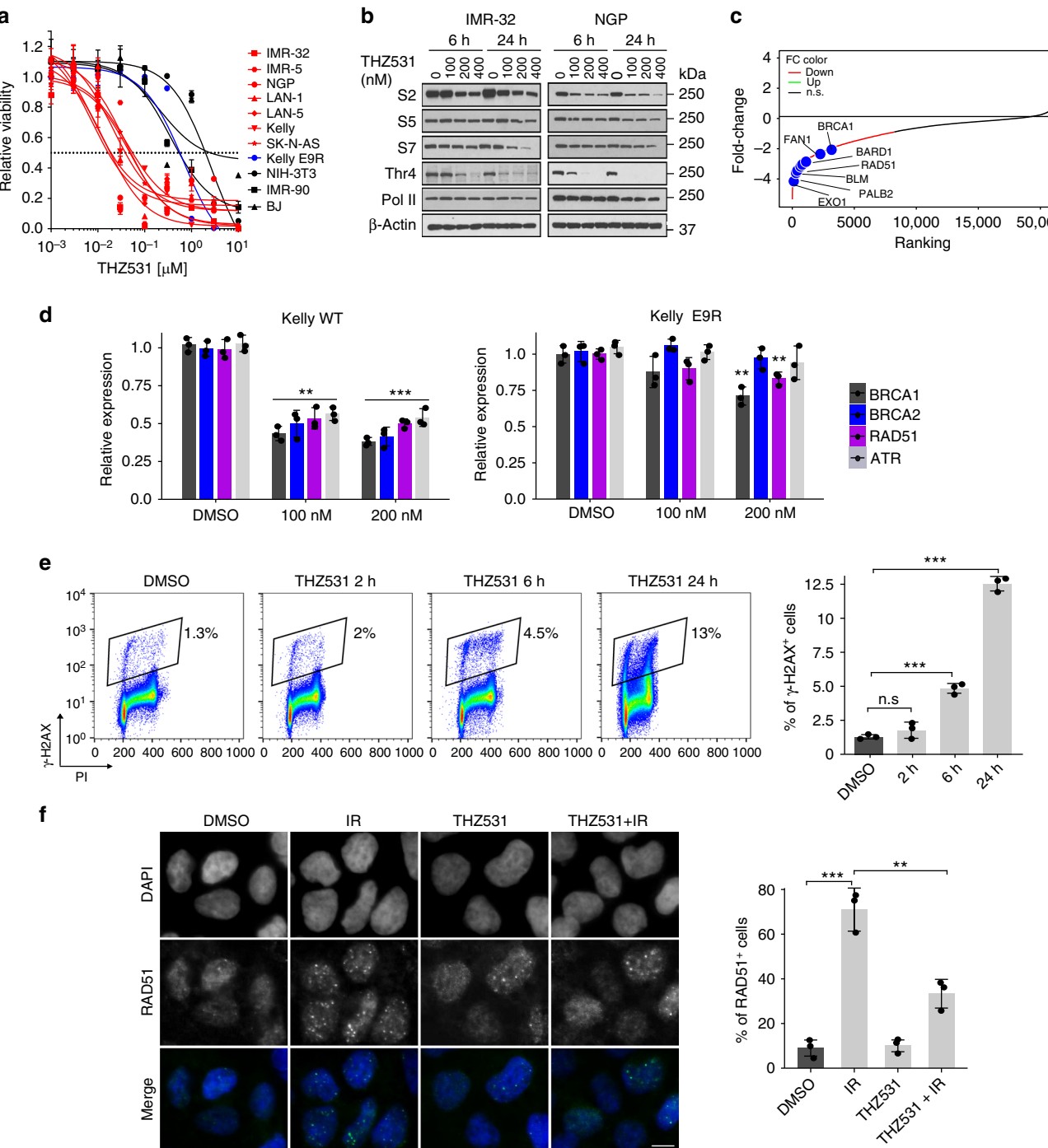

**Fig. 1** CDK12/13 inhibition results in selective cytotoxicity in NB cells and affects transcription elongation. **a** Dose-response curves for human NB cells treated with increasing concentrations of THZ531 for 72 h. Kelly E9R cells, which express a homozygous mutation at the Cys1039 THZ531-binding site in CDK12[20] (see Methods) were also included. Fibroblast cells (NIH-3T3, IMR-90, BJ) were used as controls. Cytotoxicity is reported as percent cell viability relative to DMSO-treated cells. Data represent mean ± SD; $n = 3$. **b** Western blot analysis of Pol II phosphorylation in NB cells treated with THZ531 or DMSO at the indicated concentrations for the indicated times. **c** Waterfall plot of fold-change in gene expression in IMR-32 NB cells treated with THZ531, 400 nM for 6 h; selected DDR genes are highlighted. **d** qRT-PCR analysis of the indicated DDR gene expression in Kelly WT (left) cells and Kelly E9R (right), treated with THZ531 or DMSO at the indicated concentrations for 6 h. Data are normalized to GAPDH and compared to the DMSO control. **e** Flow cytometry analysis of γ-H2AX staining in Kelly NB cells treated with 400 nM THZ531 for the indicated time points (left). Gating was performed as shown in the left panel. Numbers indicate the percentages of living cells that stained positive for γ-H2AX. Quantification of staining (right). **f** Immunofluorescence staining of RAD51 focus formation in Kelly NB cells treated with THZ531 (400 nM) or DMSO for 24 h prior to exposure to gamma radiation (IR, 8 Gy). Nuclei are stained with DAPI (scale bar, 10 μM). Quantification of staining (right) of RAD51+ cells (>5 RAD51 foci per cell). Throughout the figure, error bars indicate mean values ± SD of three independent experiments, **$p < 0.01$, ***$p < 0.001$; two-tailed Student's *t*-test

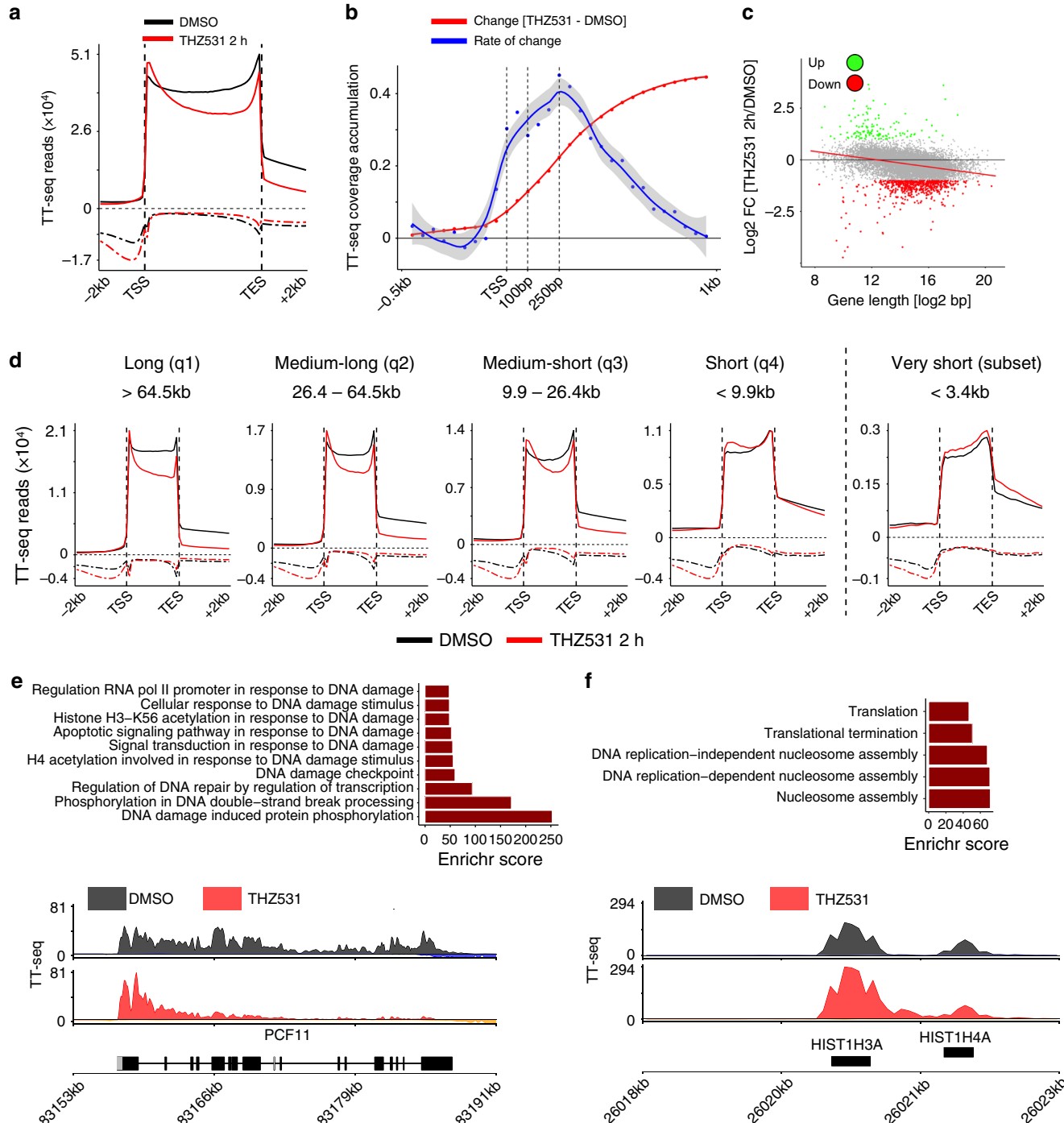

**Fig. 2** CDK12/13 inhibition leads to an elongation defect that is gene-length dependent. **a** Average metagene profiles of normalized TT-seq reads over gene bodies and extending −2 to +2 kb of all detected genes in cells treated with THZ531 400 nM for 2 h. Sense and antisense reads are depicted by solid and dashed lines respectively. **b** Average metagene profile depicting the change (red) and rate of change (blue) in TT-seq read densities in regions flanking the TSS (−0.5 to +1 kb) in cells treated with THZ531. **c** Scatter plot showing log2 fold-changes in gene expression vs. gene length in log2 scale for each protein coding gene in cells treated as in panel **a** ($R^2 = 0.12$, $p = 2e-277$, $F$-test, Spearman correlation coefficient $= -0.42$). Differentially expressed genes are indicated (FDR < 0.1 and log2 FC > 1). **d** Average metagene profiles for protein-coding genes (as in **a**) stratified according to quartiles of gene length distribution and for very short genes. Sense and antisense reads are depicted by solid and dashed lines respectively. **e** Gene ontology (GO) enrichment analysis of long genes (>64.5 kb) (top); TT-seq tracks of nascent RNA expression at the *PCF11* locus in NB cells treated with DMSO or THZ531 as in panel a (bottom). **f** GO enrichment analysis of very short genes (<3.4 kb) (top); TT-seq tracks at the *HIST1H3A/HIST1H4A* loci (bottom)

state RNA would not be sufficient to fully characterize alterations in the tightly coupled Pol II transcription and RNA processing or discriminate between early and late effects due to CDK12 perturbation. Hence, we used transient transcriptome sequencing (TT-seq), a modification of the 4-thiouridine (4sU)-pulse labeling

method[28], with spike-in controls for normalization of input RNA amount to identify the immediate changes in nascent RNA production in cells exposed to THZ531 for 30 min and 2 h. The data showed adequate exonic and intronic coverage, including that of 5′ upstream and 3′ downstream regions outside annotated

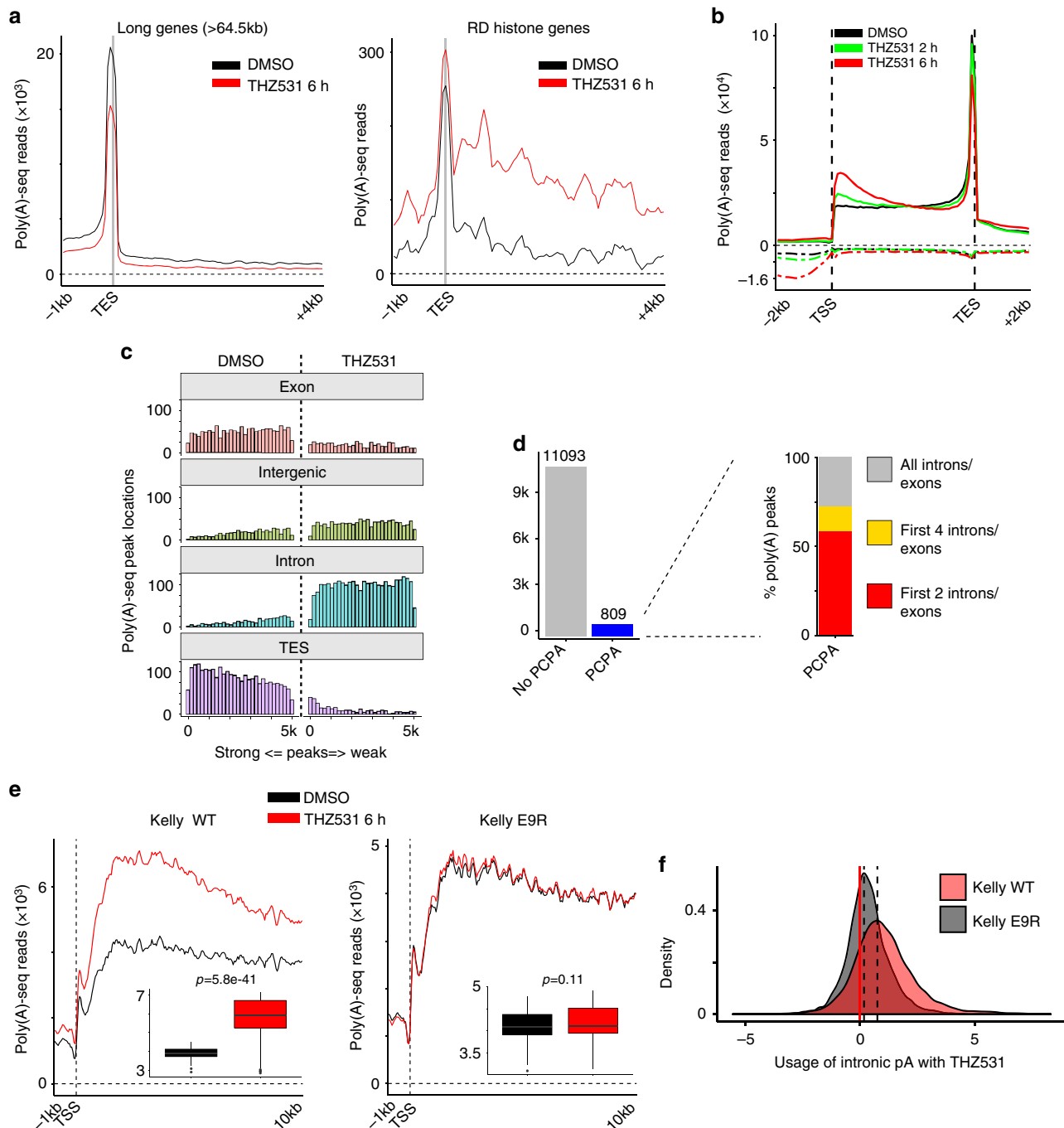

**Fig. 3** CDK12 inhibition leads to PCPA of long genes. **a** Average metagene profiles of normalized poly(A) 3′-seq reads at the transcription end sites (TES) (−1 to +4 kb) of all long genes (>64.5 kb) (left), and short genes (RD histone genes) (right). **b** Average metagene profiles of normalized poly(A) 3′-seq reads over gene bodies and extending −2 to +2 kb of all detected genes in cells treated with THZ531 400 nM for 2 and 6 h. Sense and antisense reads are depicted by solid and dashed lines, respectively. **c** Histograms showing the genomic distributions and rankings of the top 5000 poly(A) 3′-seq peaks in DMSO- and THZ531-treated cells (400 nM, 6 h). The poly(A) 3′ peaks were binned according to the depicted genomic regions and their intensities (x-axis). **d** Bar plot indicating the number of protein-coding genes that underwent premature cleavage and polyadenylation (PCPA) with THZ531. The expanded window on the right shows the genomic distribution of the identified intronic poly(A) sites. **e** Average metagene profiles of normalized poly(A) 3′-seq reads at the TSS (−1 to +10 kb) for all detected genes in Kelly WT (left) and Kelly E9R (right) cells. Changes (insets) in read density between DMSO- and THZ531 (200 nM, 6 h)-treated Kelly WT (p = 5.8e−41) and Kelly E9R (p = 0.11) cells; comparisons between groups by Wilcoxon rank-sum test. The center line indicates the median for each data set. **f** Density plot of odds-ratios of poly(A) site usage (intronic vs 3′ UTR) for genes in Kelly WT and E9R cells (p = 0, Kolmogorov-Smirnov test)

transcripts (mean 59% of reads to introns, 35% to exons and 6% to flanking regions), indicating that nascent RNAs, including a large proportion of preprocessed RNAs were captured (Supplementary Fig. 3a). Altogether, we detected 12,260 protein coding,

4809 long non-coding and 3816 short non-coding genes (transcripts per million > 2). At 30 min post-treatment, several immediate-early response gene transcripts were induced, thus confirming the ability of TT-seq to detect early changes in

transcription, but this effect was not sustained at 2 h (Supplementary Fig. 3b; Supplementary Data 2). Instead, the changes in nascent transcription were more pronounced at 2 h, leading us to focus our analyses on this time point. DDR genes were on average more downregulated compared to other genes (Supplementary Data 3; Supplementary Fig. 3c, d), consistent with our gene expression profiling of steady-state RNA, which had demonstrated downregulation of these genes after a 6-h treatment with THZ531 (Fig. 1c and Supplementary Fig. 2a–c). We validated this result by measuring nascent RNA expression along the *BRCA1* gene by qRT-PCR, observing a gradual decline in expression from the 5′ to the 3′ end of the gene following THZ531 treatment (Supplementary Fig. 3e). Gene ontology (GO) enrichment analysis of the top 400 most downregulated genes also revealed genes associated with transcription and mRNA processing (Supplementary Fig. 3f).

To further elucidate the effect of CDK12/13 inhibition on RNA synthesis, we first analyzed the changes in nascent RNA expression over gene bodies. Average meta-gene analysis of protein-coding genes and all classes of long noncoding RNAs demonstrated a prominence of TT-seq signals both upstream and downstream of the transcription start sites (TSS) (Fig. 2a). Since increased Pol II pausing has recently been shown to inhibit new transcription initiation[29,30], this result led us to ask whether pausing was affected by CDK12/13 inhibition by calculating the change in nascent transcript read density over regions flanking the TSS (−500 to 1000 bp) following THZ531 treatment. This analysis showed a gradual increase in TT-seq reads, with peak signal accumulation occurring 1000 bp downstream of the TSS—well beyond known Pol II pausing sites (20–100 bp)[31] (Fig. 2b, Supplementary Fig. 4a). Moreover, the rate at which the change in TT-seq signals occurred following THZ531 treatment, calculated by computing the difference in accumulation between consecutive 50 bp bins, continued to increase up to 250 bp beyond the TSS, after which a decrease was seen (Fig. 2b). Together, these observations suggest that THZ531 treatment does not delay Pol II pause release; in fact, in keeping with the recently proposed model[29,30], pause release may even be increased, which in turn would account for the observed increase in initiation. The finding that upstream antisense RNAs (which are short-lived and do not undergo extensive processing) were also increased at the TSS (Fig. 2a) supports this notion. After the initial 5′ increase in read density, a rapid loss of reads from the 5′-ends to the 3′-ends of genes was seen (Fig. 2a), with a net average loss of read density of around 6 kb 3′ of the TSS (Supplementary Fig. 4a). Together, these findings point to an elongation defect upon CDK12/13 inhibition.

**THZ531 induces a gene length-dependent elongation defect**. Because of the wide range in gene lengths throughout the human genome (<1 kb to >1 Mb) and prior reports that CDK12 preferentially regulates the expression of long genes[6], we next determined whether this variable had any effect on the elongation defect seen with the CDK12/13 inhibitor. Notably, there was a significant correlation between gene length and downregulation of gene expression: the longer the gene, the more likely it was to be downregulated (Fig. 2c). To define this relationship further, we divided the downregulated genes into 4 quartiles based on the distribution of gene lengths [short (<9.9 kb), medium-short (9.9–26.4 kb), medium-long (26.4–64.5 kb) and long (>64.5 kb)]. As shown in Fig. 2d, the long genes consistently had the most pronounced elongation defect and, concomitantly, the greatest transcriptional downregulation. When we restricted our differential gene expression analysis to protein-coding genes and nascent RNA reads that fell within exonic regions, and compared the

results against these unbiased gene length groups, we observed that 362 (7%) of 5110 longer genes (202 long and 160 medium-long) were downregulated, while only 111 (2%) of 4895 shorter genes (14 medium-short and 97 short) were upregulated (adjusted $p < 0.05$; log2 fold change <−1). GO analysis of these genes showed that the top categories comprised DDR genes (Fig. 2e). This length-dependent elongation defect was not observed for long or short noncoding RNAs, although global downregulation of the latter was observed (Supplementary Fig. 4b). In contrast to longer genes, we observed that short or very-short (<3.4 kb) genes were upregulated or transcribed normally and showed significant 3′ UTR increase and extension (Supplementary Fig. 4c). This subgroup was enriched for replication-dependent (RD) histone genes ($n = 72$) (Fig. 2f) which do not normally rely on polyadenylation for transcription termination but on stem-loop binding[32,33], a process regulated by Pol II Thr4P that was profoundly decreased following THZ531 treatment (Supplementary Fig. 4d, Fig. 1b). This effect (validated by qRT-PCR of total and nascent RNA) was also prominent in the amplified and overexpressed *MYCN* oncogene (6.4 kb long) and explained its lack of downregulation in *MYCN*-amplified cells treated with THZ531 (Supplementary Fig 4e, f). Together, these results suggest that CDK12/13 inhibition with THZ531 leads to an elongation defect which predominantly involves genes within the longer length categories, with normal or increased expression of shorter genes.

**CDK12/13 inhibition leads to PCPA**. The gradual decrease in nascent RNA reads from the 5′ to the 3′ ends of long genes with THZ531 treatment implied a possible termination defect. To pursue this notion, we performed global poly(A) 3′-sequencing of cells treated with THZ531. The majority (69%) of the identified poly(A) peaks were associated with known upstream polyadenylation site (PAS) motifs, the most abundant being the canonical AATAAA motif (Supplementary Fig. 5a). In agreement with their nascent RNA expression profiles (Fig. 2e, f), transcripts associated with long genes showed a loss of annotated terminal or 3′ poly(A) sites (Fig. 3a, left), while short transcripts, such as those of the histone processing genes and *MYCN*, terminated at distal unannotated poly(A) sites (Fig. 3a, right, Supplementary Fig. 4e). These findings suggest a transcription termination defect following CDK12/13 inhibition, with differential effects based on gene length.

Given the decrease in annotated terminal poly(A) sites, we attributed the genome-wide elongation and termination defects in THZ531-treated cells to premature cleavage and polyadenylation (PCPA). Transcription of most protein-coding genes is terminated at the 3′ ends of genes through cleavage and polyadenylation, whereas premature transcription termination such as PCPA occurs within a short distance from the TSS and results in the production of truncated transcripts[34–36]. We therefore analyzed the distribution of poly(A) 3′-seq reads for all protein-coding genes across the genome, observing a time-dependent increase in polyadenylated sites at the 5′ proximal ends of genes following CDK12/13 inhibition (Fig. 3b). Further study of the poly(A) sites that were differentially utilized between THZ531- and DMSO-treated cells focused on those that showed at least a 2-fold change (Supplementary Fig. 5b). This approach revealed strikingly different 3′-peak distributions in THZ531-treated compared with DMSO-treated cells, with significant enrichment in intronic regions (60% vs. 36%) and an almost complete absence of reads at annotated transcription end sites (TES) (4% in THZ531-treated vs. 20% in DMSO-treated cells; Supplementary Fig. 5c), an effect that was most prominent for the top 5000 differential poly(A) peaks between the two samples (Fig. 3c). Together, these findings indicate that CDK12/13 inhibition in NB cells causes generalized

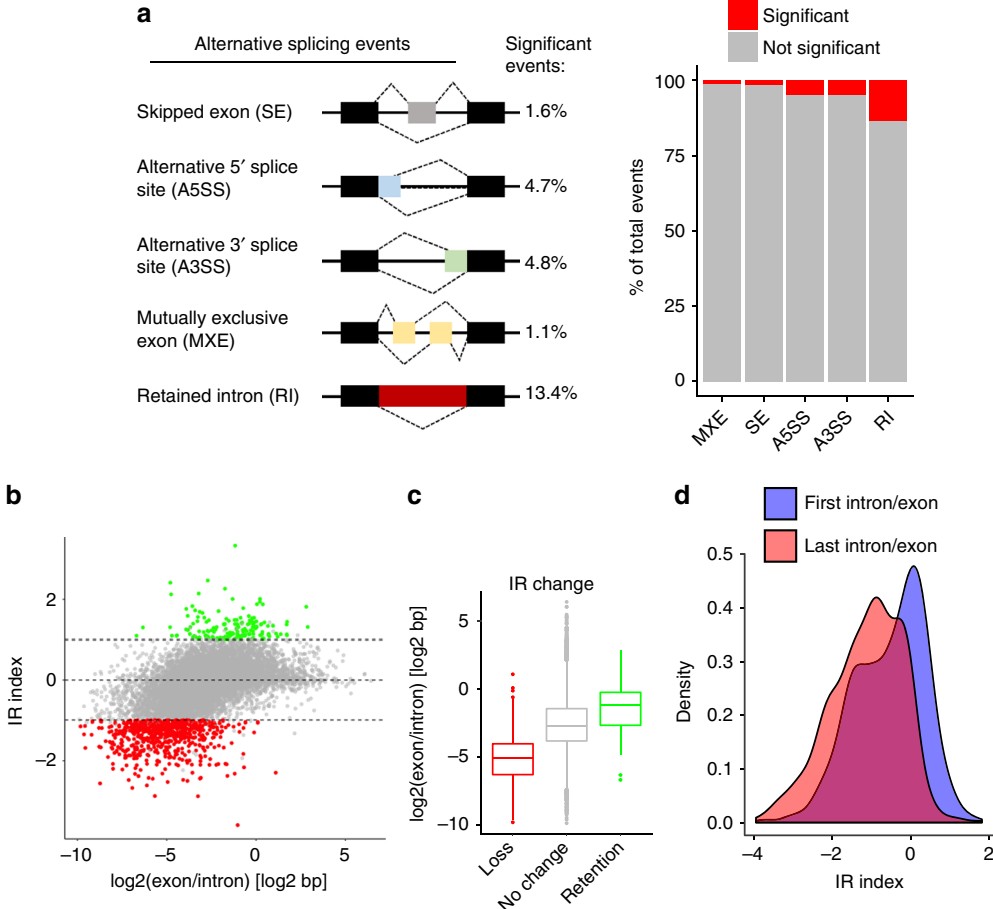

**Fig. 4** CDK12/13 inhibition results in minimal splicing alterations. **a** Diagrammatic representation (left) and bar plot of splicing events (right) observed in TT-seq analysis of NB cells treated with THZ531 (400 nM) for 2 h. **b** Scatterplot of intron retention index (IR index) vs. the ratio of exon and intron lengths in log2 scale. Genes with an IR index >1 or ≤1 display intron retention and loss respectively (adjusted $p < 0.05$, Fisher's exact test). **c** Box plot illustrating the length distributions of genes that display intron loss or retention. The center line indicates the median for each data set. **d** Density plots illustrating the contributions of the proximal (first intron/exon) and distal (last intron/exon) gene regions in calculation of the IR index. Comparison of IR index distribution between proximal and distal intron/exon pairs ($p = 0$, Kolmogorov–Smirnov test)

PCPA with the use of cryptic intronic polyadenylation sites. Interestingly, the THZ531-induced effect at the nascent RNA level was computationally inferred[37] as occurring as early as 2 h post-treatment, with PCPA apparent in 809 (7%) of the 11,902 protein-coding genes containing at least one intron (Fig. 3d). Poly (A) 3′-seq data showed that more than half of these genes underwent early termination in the first two introns/exons (59%, 476/809) and almost three-quarters in the first four introns/exons (73%, 587/809) (Fig. 3d). Integrative analysis of TT-seq and poly (A) 3′-seq data at the 5′ proximal regions (−1 to +1 kb of TSS) revealed that the aberrant accumulation of 5′ proximal TT-seq reads coincided with the peaks of proximal 3′ poly(A) usage, implying that most transcripts were terminated early at the beginning of productive elongation (Supplementary Fig. 5d). This conclusion was further supported by the inverse correlation between nascent reads along the 5′–3′ regions and the usage of proximal poly(A) sites in THZ531-treated cells, suggesting a high probability of proximal poly(A) site usage that gradually diminishes when elongation is terminated due to PCPA.

**The THZ531-induced termination defect is due to CDK12 loss**. We next asked whether the observed effects on termination through PCPA could be assigned specifically to CDK12 or 13 by genetic depletion (shRNA KD) followed by poly(A) 3′-sequencing. CDK12-depleted cells displayed the highest and most

significant increase in poly(A) 3′-sequencing reads at the 5′ proximal ends of genes compared to control shRNA-expressing cells (Supplementary Fig 5e). Although depletion of CDK13 also resulted in an increase in 5′ proximal reads, this effect was significantly lower than that seen with CDK12 depletion (Supplementary Fig 5e). Only CDK12-depleted cells showed an increased usage of intronic poly(A) sites; this phenomenon was not evident in CDK13-depleted cells (Supplementary Fig 5f). Importantly, THZ531 treatment in Kelly E9R cells with the THZ531-binding site mutation did not display any increase in 5′ proximal reads (Fig. 3e) or in intronic poly(A) site usage compared to wild-type Kelly cells (Fig. 3f), suggesting that targeting of CDK13 alone was not sufficient to induce the PCPA defect. The gene length-dependent decrease in nascent RNA expression observed following THZ531 treatment (Fig. 2c) was also noted in cells with CDK12 shRNA depletion, but not in cells with CDK13 shRNA KD or in E9R cells treated with THZ531 (Supplementary Fig. 5g). Together, these results further identify PCPA as the main defect resulting from THZ531 treatment, an outcome that is mediated primarily by its targeting of CDK12.

**CDK12/13 inhibition induces minimal splicing alterations**. Because previous studies point to a role for CDK12 in splicing regulation[27,38], we determined whether aberrant splicing could explain the elongation defect seen with THZ531 treatment.

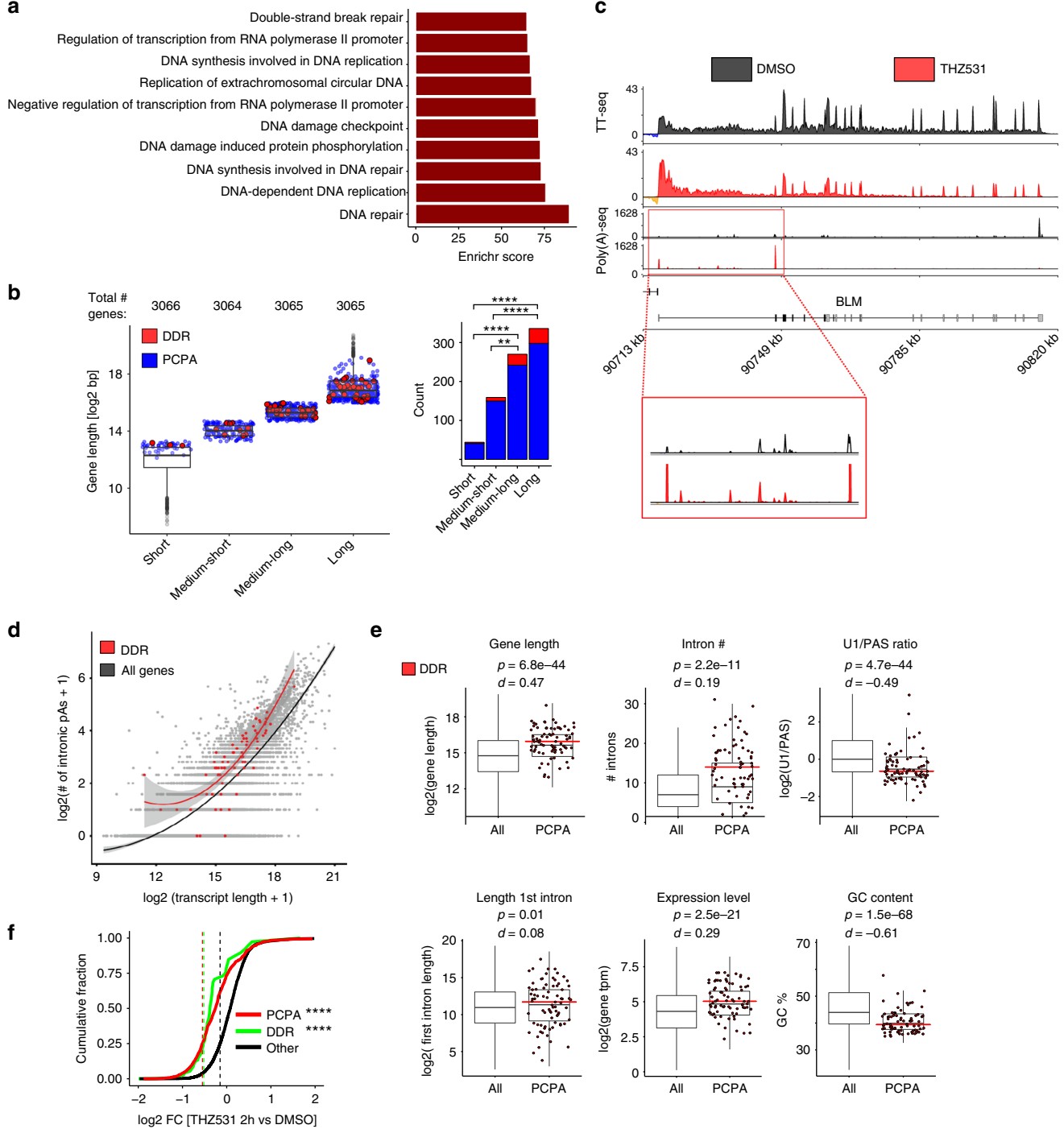

**Fig. 5** Gene length and a lower U1/PAS ratio predispose DDR genes to PCPA. **a** GO enrichment analysis of the 809 genes that underwent PCPA (FDR < 0.01) based on TT-seq analysis of cells treated with THZ531 (400 nM for 2 h). **b** Box plots and bar plots showing the distribution and numbers of PCPA and DDR genes in the different gene-length categories established in Fig. 2d (****$p$ < 0.0001, **$p$ < 0.01, Fisher's exact test). The center line indicates the median for each data set. **c** TT-seq and poly(A) 3′-seq tracks at the *BLM* DDR gene locus depicting the loss of annotated terminal polyadenylation signal and the presence of early termination due to PCPA in cells treated with THZ531 as in **a**. **d** Number of intronic poly(A) sites as a function of transcript length. A polynomial regression curve is plotted for all genes (black) and DDR genes only (red) ($p$ = 1.7e−13, predicted vs. observed, Wilcoxon rank-sum test). **e** Box plots comparing the indicated determinants of PCPA in all genes vs. PCPA genes only and the proportion of DDR genes within the latter subset (see figure for $p$ and $d$ values; Wilcoxon rank-sum test & Cohen's $d$ effect-size, respectively). The black and red center lines indicate the median of all PCPA and DDR genes respectively. **f** Cumulative fraction plot showing the change in expression of PCPA ($p$ = 2.2e−16, Kolmogorov-Smirnov test) and DDR ($p$ = 1.9e−14, Kolmogorov-Smirnov test) transcripts relative to other transcripts following THZ531 treatment as in **a**

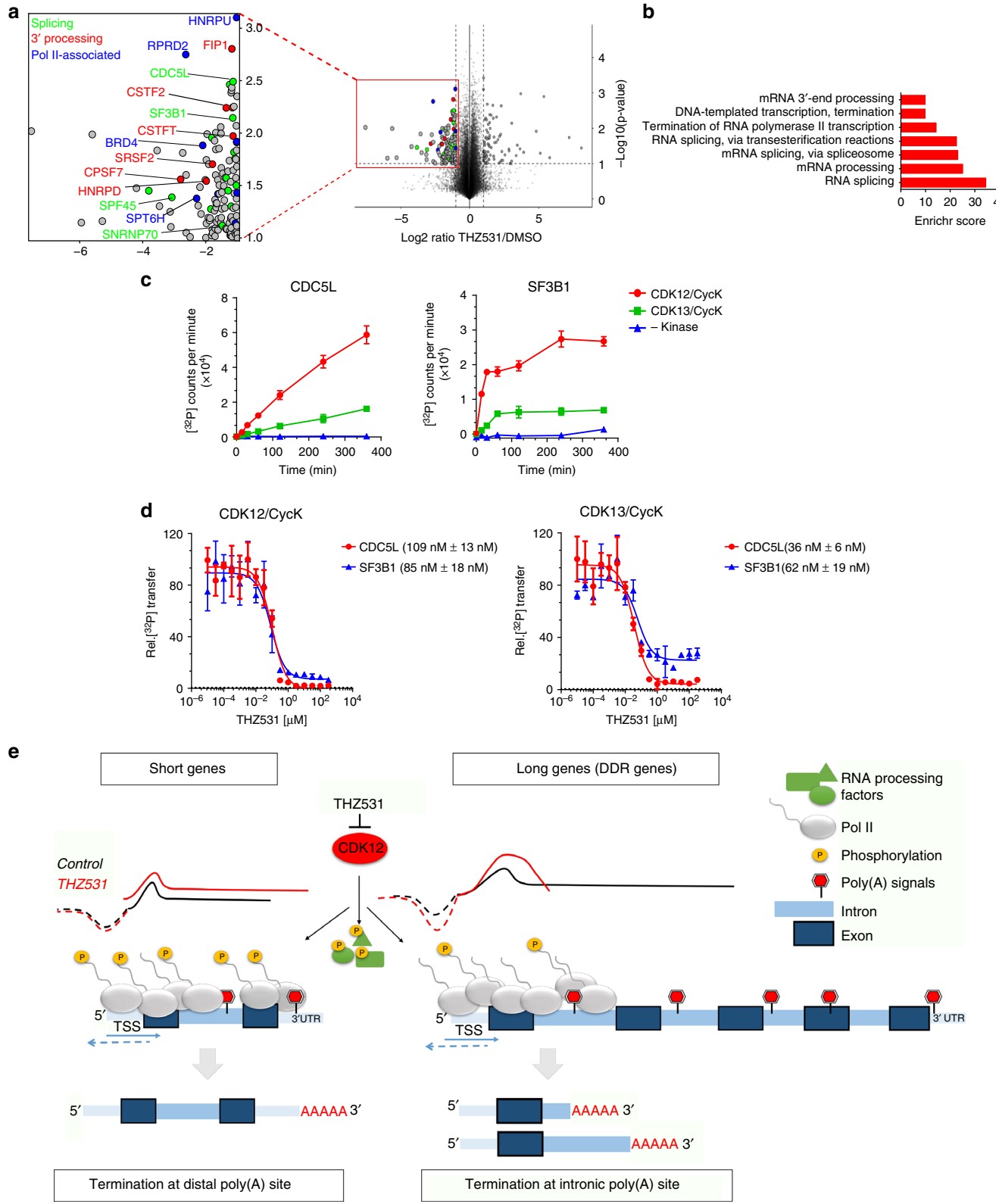

Analysis of the nascent transcriptomic data showed that in general, there was a paucity of significantly altered splicing events following THZ531 treatment. The largest proportion of splicing defects comprised intron retention (13.4%), followed by alternative 5′ and 3′ splicing (4.7% and 4.8% respectively), while skipped and mutually exclusive exons were rarely observed (Fig. 4a). To further investigate intron retention, we calculated the intron retention (IR) index (log2 ratio of intron vs. exon TT-seq signal coverage differences between THZ531- and DMSO-treated cells; see Methods), and noted overall intron loss (642 of 11,155 protein-coding genes, 5.7%) together with a low exon/intron length ratio (IR < 1) (Fig. 4b) in genes that were downregulated by THZ531, in fact, suggestive of increased splicing efficiency. Importantly, this effect was seen primarily at long genes. Short

**Fig. 6** CDK12/13 phosphorylates RNA processing proteins. **a** Volcano plot of proteome-wide changes in phosphorylation site occupancy identified through SILAC analysis of NB cells treated with THZ531, 400 nM for 2 h. Expanded box shows selected co-transcriptional RNA processing proteins. **b** GO terms for candidate CDK12/13 substrates. **c** In vitro kinase assays of CDK12/CycK (red)-mediated and CDK13/CycK (green)-mediated phosphorylation of CDC5L (aa 370–505) and GST-SF3B1 (aa 113–462) at the indicated time points. A negative control measurement without kinase is shown in blue. Radioactive kinase reactions were performed with 0.2 μM CDK12/CycK or CDK13/CycK and 50 μM substrate protein, respectively. Data are reported as mean ± SD, $n = 3$. **d** Dose-response curves of THZ531 incubated with recombinant CDK12 (left) and CDK13 (right) protein and CDC5L (aa 370–505) and GST-SF3B1 (aa 113–462). Radioactive kinase reactions were performed after 30 min preincubation with increasing concentrations of THZ531. For all incubation time series, the counts per minute of the kinase activity measurements were normalized to the relative $^{32}$P transfer. Data are reported as mean ± SD; $n = 3$. IC$_{50}$ values shown in parentheses. **e** Model of CDK12 as a regulator of pre-mRNA processing. CDK12 phosphorylates and thus likely stimulates the orchestrated action of RNA Pol II CTD and RNA processing proteins. CDK12 inhibition leads to a gene-length-dependent productive elongation defect associated with early termination through premature cleavage and polyadenylation (PCPA). Especially vulnerable to PCPA are long genes with a lower ratio of U1 snRNP binding to poly(A) sites, which include many of those involved in the DDR. Among short genes, including genes that normally terminate through stem-loop binding, CDK12 inhibition increases intron retention and leads to longer polyadenylated transcripts

---

genes, on the other hand, were characterized by intron retention (156 of 11,155 genes, 1.3%) and a high exon/intron length ratio (IR > 1, adjusted $p < 0.05$) (Fig. 4c). We reasoned that the apparent increased splicing efficiency in long genes was likely not due to a more efficient spliceosome, but rather, a secondary effect of the severe elongation defect seen within these genes (Fig. 2a, d). To pursue this hypothesis, we calculated the individual IR indices for the combination of the first exon/intron and last exon/intron length-ratios of the long genes that displayed intron loss, observing a greater intron loss for the last exon/intron compared to that of the first exon/intron (Fig. 4d). These results suggest that the lack of intron coverage at the 3′ end in longer genes was likely due to defective elongation together with the reduced formation of such long transcripts following THZ531 treatment.

**Gene length and the U1 snRNP/PAS ratio influence PCPA.** Genes that underwent THZ531-induced PCPA were significantly longer than genes that did not undergo this change, as might be expected from the elongation defect in the long gene group (>64.5 kb; Fig. 2d, Supplementary Fig. 6a). Importantly, the group of long genes that underwent PCPA was specifically enriched for DDR genes, such as *BARD1* and *BLM*, with respective lengths of 84 and 98 kb (Fig. 5a–c, Supplementary Fig. 6b). We validated this finding through 3′ RACE of the *BARD1* transcript in THZ531-treated cells (Supplementary Fig. 6c). Interestingly, we noted that DDR genes undergoing PCPA as a result of CDK12 inhibition had a statistically higher number of intronic poly(A) sites relative to other genes of similar length (Fig. 5d), indicating that gene length alone does not fully explain the specific vulnerability of this subset of genes to early termination. Hence, to assess the relative contribution of gene length to the early termination phenotype observed after THZ531 treatment, we tested other determinants known to influence co-transcriptional processing[36,39,40]. Apart from longer gene length, we noted that a longer first intron, a larger number of introns, higher gene expression, lower GC content and a lower U1 snRNP/PAS ratio were also associated with early termination due to PCPA, with the latter two features emerging as the most significant based on effect size (Fig. 5d, e).

The U1 snRNP complex prevents premature termination through recognition and inhibition of cryptic poly(A) sites[35–37,41]. Indeed, Oh et al.[37] demonstrated that direct depletion of U1 in HeLa cells using morpholino KD results in the decreased expression of long genes. We observed a significant overlap between genes that underwent PCPA in this data set and those that were similarly affected by THZ531 treatment, even though they represent two different cancer cell types and were studied at different time points after perturbation of different targets – U1 at 4 and 8 h[37] and CDK12 at 2 h (this study) (Supplementary Fig. 7a; Supplementary Data 4). This finding is supported by the significantly increased

usage of intronic poly(A) sites in DDR genes, even when compared with the genome-wide increase that was observed following THZ531 treatment in wild-type Kelly NB cells (Supplementary Fig. 7b, left; Fig. 3f). Importantly, no such change was seen in Kelly E9R THZ531-resistant cells (Supplementary Fig. 7b, right). In addition, genes that showed increased intronic poly(A) site usage following THZ531 exposure were enriched for GO categories associated with DNA damage (Supplementary Fig. 7c), and their expression was significantly reduced in WT compared to E9R cells expressing the Cys1039 mutation (Supplementary Fig. 7d, e). In conclusion, these observations indicate that CDK12 inhibition leads to premature termination that depends on gene length and the U1 snRNP/PAS ratio and may provide an explanation for the selective effects of this transcriptional kinase on DDR gene expression (Fig. 5f).

**CDK12/13 phosphorylates RNA processing proteins.** Our results demonstrate the effect of CDK12 inhibition on transcription elongation and identify PCPA as a potential explanation for this selectivity. Given that the transcriptional activity of Pol II and processing of nascent transcripts occur simultaneously[2], we hypothesized that the CDK12 and/or 13 kinases may regulate the phosphorylation of targets other than the Pol II CTD, and could contribute to cotranscriptional RNA processing. To address this question, we performed phosphoproteomics analyses of cells treated with and without THZ531 using stable isotope labeling with amino acids in cell culture (SILAC). This study revealed a ≥2-fold increase of 88 phosphopeptides and a similar decrease in 129 sites ($p < 0.1$; Student's $t$-test; Fig. 6a, Supplementary Data 5). The majority of phosphorylation sites that decreased in abundance upon THZ531 treatment occurred at serine or threonine residues, usually with a proline in the +1 position—the minimal consensus recognition site for all CDKs[42] (Supplementary Fig. 8a). Protein interaction network analysis of all identified substrates clustered into two groups, the larger of which contained phosphorylated proteins centered on Pol II, while the other consisted of phosphorylated proteins that interact directly with CDK12 (Supplementary Fig. 8b). Interestingly, proteins encoded by DDR genes were not significantly represented in this analysis, suggesting that CDK12 may not directly regulate DDR protein phosphorylation. GO analysis of candidate CDK12 substrates that were significantly decreased in abundance after THZ531 treatment revealed mRNA processing factors as the top category, accounting for more than 50% of the identified phosphoproteins (Fig. 6b). Interestingly, one of the top mRNA processing factors was the small nuclear ribonucleoprotein SNRNP70, which associates with U1 as part of the U1 snRNP complex[43] (Fig. 6a). Other top phosphoproteins that were affected by CDK12/13 inhibition included the PRP19 complex protein[44,45], CDC5L with roles in RNA splicing and genomic stability and SF3B1, a

component of the splicing machinery that is involved in pre-mRNA splicing[46]. We confirmed the phosphorylation of these candidates using $^{32}$P-labeled ATP in vitro kinase assays using GST-tagged substrates together with CDK12/CycK and CDK13/CycK (Supplementary Fig. 8c). Similar to CDK12, CDK13 phosphorylated the substrate proteins in a time-dependent manner (Fig. 6c). Of note, CDK12-mediated phosphorylation resulted in a higher rate of $^{32}$P incorporation for CDC5L and SF3B1, suggesting that CDK12 phosphorylates more sites in these substrates than CDK13 (Fig. 6c). Additionally, control experiments without the addition of either kinase revealed that phosphorylation of CDC5L and SF3B1 was significantly below that measured in presence of the active kinases. Next, we repeated the kinase assays after pre-treatment of the CDK/cyclin complex with THZ531, noting reduced phosphorylation of the CDC5L and SF3B1 substrate proteins with increasing concentrations of the inhibitor (Fig. 6d). Finally, to identify the exact sites phosphorylated by CDK12/CycK in the in vitro kinase assays, we performed peptide mass fingerprint analyses of the recombinant protein substrates, which confirmed the following phosphorylation sites identified in the SILAC analysis: CDC5L (pT396), SF3B1 (pT326), (Supplementary Fig. 8d, Supplementary Data 6). Together, these results suggest that both CDK12 and 13 phosphorylate pre-mRNA processing factors that could affect their recruitment to Pol II.

## Discussion

In this study, we took advantage of the selectivity and irreversibility of a covalent inhibitor of CDK12/13 to dissect the early alterations in cotranscriptional RNA processing in NB cells. Using nascent RNA and poly(A) 3′-sequencing, we demonstrate that such inhibition leads to a gene length-dependent elongation defect associated with early termination through PCPA (Fig. 6e). Especially vulnerable to this defect were long genes with a lower ratio of U1 snRNP binding to poly(A) sites, which include many of those involved in the DDR. Conversely, short genes showed an increased likelihood of intron retention and 3′ UTR extension or, as in the case of the non-polyadenylated replication-dependent histone genes, the generation of polyadenylated transcripts. We further identified CDK12 as the predominant kinase mediating the transcriptional effects of THZ531 in treated cells. NB cells harboring a point mutation at the CDK12 Cys1039 binding site of THZ531 were less sensitive to the inhibitor and had significantly fewer length-dependent elongation defects and PCPA, compared to findings in cells expressing WT CDK12. The distinction among phosphorylation targets was not as clear-cut; both CDK12 and CDK13 induced the phosphorylation of RNA processing proteins, with further studies needed to resolve this overlap.

The CDK12-mediated transcriptional effects reported here agree with—but differ mechanistically from—those reported with inhibition of the other Pol II Ser2 elongation kinase, CDK9, where increased Pol II pausing leads to a defect in elongation and negatively impacts transcription initiation[29,30]. By contrast, our data show that perturbation of CDK12, while also resulting in an elongation defect, is likely to be associated with increased transcription initiation and Pol II pause release. This conclusion is supported by the accumulation of nascent RNA reads both upstream (antisense) and at the TSS, extending well beyond the Pol II pause sites to average peak densities ~+1000 bp downstream of the TSS, and especially by the continued rate of increase in signal accumulation beyond the pause sites. Thus, given that the accumulation of TT-seq reads coincided with the onset of productive elongation during which Pol II accelerates dramatically[47,48], it is likely that CDK12 function is critical for the recruitment and/or modification of components of the

transcription machinery that together sustain efficient rates of productive elongation[49]. Alternatively, or concomitantly, the accumulation of nascent RNA reads downstream of the TSS could indicate the existence of CDK12-dependent elongation checkpoints similar to those reported for CDK9[50]. We also observed a sharp decrease in TT-seq reads +1000 bp downstream of the TSS, with resultant early termination, a finding supported by increased poly(A) 3′-seq reads at the 5′ proximal ends of these genes. Thus, we postulate that in the absence of CDK12 activity, Pol II is less capable of entering into productive elongation; instead, as previously proposed[51], it is gradually released from chromatin, likely increasing the pool of free Pol II molecules that can engage in transcription initiation, and accounting for the increased TT-seq reads at the TSS in THZ531-treated cells.

We observed that in addition to gene length, a main determinant of premature termination was the U1 snRNP/PAS ratio, which was lower in DDR genes that underwent PCPA. It is well established that the U1 snRNP facilitates the transcription of long genes with its inhibition resulting in PCPA[37]. SNRNP70, a component of the U1 snRNP complex, was identified as a potential phosphorylation substrate of CDK12/13 in our study; hence, it is quite possible that its decreased phosphorylation could partly account for the increased usage of alternate polyadenylation sites in DDR and other long genes. This could also explain why in contrast to findings in other studies implicating CDK12 in splicing regulation[27,38], CDK12 inhibition did not lead to major splicing alterations, most likely because transcription was terminated well before it reached the 3′ splice sites.

As shown schematically in Fig. 6e, we propose that CDK12 inhibition leads to an increased probability of using cryptic intronic poly(A) sites and undergoing PCPA, possibly due to a slowing of productive Pol II elongation. As such, long genes with low U1 snRNP/PAS ratios, such as DDR genes, are especially vulnerable to this loss, yielding an aborted elongation phenotype, manifested at the 3′ ends of these genes. Most importantly, our analysis demonstrates that CDK12 by itself lacks any intrinsic preference for DDR genes; instead, the structural properties of the gene target determine its sensitivity to CDK12 inhibition, and many DDR genes possess the requisite features. Not only was gene length a significant contributor to the PCPA phenotype, but the DDR genes significantly affected by CDK12 inhibition harbored more intronic poly(A) sites than expected based on their longer gene lengths. DDR genes that evaded PCPA were those with genetic determinants that did not favor this process—such as shorter lengths, a short first intron and decreased numbers of introns. Future work is needed to resolve why so many genes involved in DNA repair have this genetic composition compared to the genome-wide background.

In conclusion, by inducing an RNA Pol II elongation defect and subsequent usage of proximal poly(A) sites that led to premature cleavage and polyadenylation of long DDR genes, we were able to clarify the mechanism by which THZ531 selectively abolishes the DDR in NB cells, which are highly dependent on adequate DNA repair function for their survival. Dubbury et al.[52], recently examined the later effects of CDK12 depletion on total RNA expression in mouse ES cells, showing that CDK12 suppresses intronic polyadenylation as a mode of DDR gene regulation. The authors found that this mechanism is preserved in ovarian and prostate tumors with CDK12 loss-of-function mutations or deletions. Our findings augment those of the Dubbury study by (i) linking the unique susceptibility of DDR genes to CDK12 inhibition with their relatively longer lengths, lower GC content and lower ratios of U1 snRNP binding sites to intronic polyadenylation sites, and (ii) showing that these transcriptional effects occur at the nascent RNA level as early as 2 h after CDK12 loss. Thus, effects attributed to CDK12 loss do not

appear to be restricted to cancers with loss-of-function mutations, but encompass those with severe underlying DNA damage, such as NB. Moreover, as recently demonstrated in a subset of prostate cancers with CDK12 loss-of-function mutations[53], the PCPA, as well as intron retention, observed with CDK12 inhibition could facilitate the formation of neoantigens that might be exploited to improve immune therapies or to develop personalized cancer vaccines[54]. The extent to which these observations apply to other genomically unstable cancers lacking CDK12 loss-of-function mutations will be pivotal in generating molecular rationales for the therapeutic targeting of CDK12 across a broad cross-section of vulnerable tumors.

## Methods

**Cell culture**. Human neuroblastoma (NB) cells (Kelly, IMR-32, IMR-5, LAN-1, LAN-5, NGP, SK-N-AS, SH-SY5Y, CHLA-20, CHLA-15, and SK-N-FI) were obtained from the Children's Oncology Group cell line bank and genotyped at the DFCI Core Facility. The cell lines were authenticated through STR analyses. The Kelly E9R NB cell line harbors a single point mutation in CDK12 at the cysteine 1039 covalent binding site of THZ531. Specifically, this mutation was acquired spontaneously in Kelly NB cells upon exposure to escalating doses of CDK12 inhibitor, E9 over the course of few months as previously reported[20]. Human lung (IMR-90) and skin fibroblasts (BJ) were kindly provided by Dr. Richard Gregory (Boston Children's Hospital). NIH3T3 cells were purchased from the American Type Culture Collection (ATCC). NB cells were grown in RPMI (Invitrogen) supplemented with 10% FBS and 1% penicillin/streptomycin (Invitrogen). IMR-90, BJ, and NIH3T3 cells were grown in DMEM (Invitrogen) supplemented with 10% FBS and 1% penicillin/streptomycin. All cell lines were routinely tested for mycoplasma.

**Compounds**. THZ531 was prepared by Dr. Nathanael Gray's laboratory[14].

**Cell viability assay**. Cells were plated in 96-well plates at a seeding density of $4 \times 10^3$ cells/well. After 24 h, cells were treated with increasing concentrations of THZ531 (10 nM to 10 μM). DMSO solvent without compound served as a negative control. After 72 h incubation, cells were analyzed for viability using the CellTiter-Glo Luminescent Cell Viability Assay (Promega) according to the manufacturer's instructions. All proliferation assays were performed in biological triplicates and error bars represent mean ± SD. Drug concentrations that inhibited 50% of cell growth ($IC_{50}$) were determined using a nonlinear regression curve fit using GraphPad Prism 6 software.

**Fluorescence-activated cell sorting analysis (FACS)**. For cell cycle and DNA damage analysis, cells were treated with DMSO or THZ531, 400 nM. After 2, 6, and 24 h, cells were trypsinized and fixed in ice-cold 70% ethanol overnight at −20 °C. After washing with ice-cold phosphate-buffered saline (PBS), cells were incubated in PBS-0.5% Tween-20 with γ-H2AX antibody overnight at 4 °C. Cells were subsequently washed and incubated with Alexa-488-conjugated secondary antibody for 45 min and then treated with 0.5 mg/ml RNAse A (Sigma-Aldrich) in combination with 50 μg/ml propidium iodide (PI, BD Biosciences). For apoptosis analysis cells were harvested and stained with PI and FITC-Annexin V according to the manufacturer's protocol (BD Biosciences). All FACS samples were analyzed on a FACS-Calibur (Becton Dickinson) using Cell Quest software (Becton Dickinson). A minimum of 50,000 events was counted per sample and used for further analysis. Data were analyzed using FlowJo software.

**shRNA Knockdown**. pLKO.1 plasmids containing shRNA sequences targeting CDK12 (sh#1: TRCN0000001795; sh#2 TRCN0000197022), CDK13 (sh#1: TRCN0000000701; sh#2: TRCN0000000704) and GFP were obtained from the RNAi Consortium of the Broad Institute (Broad Institute, Cambridge, MA), knockdowns were performed as described previously[16]. Briefly, the constructs were transfected into HEK293T cells with helper plasmids: pCMV-dR8.91 and pMD2.G-VSV-G for virus production. Cells were then transduced with virus, followed by puromycin selection for two days.

**Western Blotting**. Cells were collected by trypsinization and lysed at 4 °C in NP40 buffer (Invitrogen) supplemented with complete protease inhibitor cocktail (Roche), PhosSTOP phosphatase inhibitor cocktail (Roche) and PMSF (1 mM). Protein concentrations were determined with the Biorad DC protein assay kit (Bio-Rad). Whole cell protein lysates were resolved on 4–12% Bis-Tris gels (Invitrogen) and transferred to nitrocellulose membranes (Bio-Rad). After blocking nonspecific binding sites for 1 h using 5% dry milk (Sigma) in Tris-buffered saline (TBS) supplemented with 0.2% Tween-20 (TBS-T), membranes were incubated overnight with primary antibody at 4 °C. Chemiluminescent detection was performed with the appropriate secondary antibodies and developed using Genemate Blue ultra-

autoradiography film (VWR). Uncropped versions of all western blots can be found in Supplementary Fig. 9.

**Immunofluorescence microscopy**. Cells were seeded on glass coverslips in six-well plates at a seeding density of $1 \times 10^6$ cells/well. After 24 h, cells were treated with DMSO or 400 nM of THZ531 for 24 h. Additionally, for the RAD51 staining, cells were irradiated (8 Gy) using a γ-cell 40 irradiator with a cesium source (Best Theratronics, Ltd). Six hours after irradiation cells were washed in PBS and fixed in 3.7% formaldehyde in PBS for 15 min at room temperature (RT). Cells were permeabilized in 0.1% Triton X-100 in PBS for 5 min. Subsequently, cells were extensively washed and incubated with PBS containing 0.05% Tween-20 and 5% BSA (PBS-Tween-BSA) for 1 h to block nonspecific binding. Cells were then incubated overnight at 4 °C with anti-RAD51 primary antibody in PBS-Tween-BSA, extensively washed and incubated for 45 min with AlexaFluor 488-conjugated secondary antibody and counterstained with DAPI. Images were acquired on a Zeiss AXIO Imager Z1 fluorescence microscope using a ×63 immersion objective, equipped with AxioVision software. Nuclei with >5 RAD51 foci were considered positive and 100 nuclei per condition were analyzed.

**Target engagement assay**. Cells were treated with THZ531 or DMSO for 6 h at the indicated doses. Subsequently, total cell lysates were prepared as for western blotting. To IP CDK12 and CDK13, 1 mg and 4 mg, respectively of total protein was incubated with 1 μM of biotin-THZ531 at 4 °C overnight. Subsequently, lysates were incubated with streptavidin agarose (30 μl) for 2 h at 4 °C. Agarose beads were washed 3x with cell lysis buffer and boiled for 10 min in 2× gel loading buffer. Proteins were resolved by WB. Fifty microgram of total protein was used as a loading control.

**Stable isotope labeling by amino acids in cell culture (SILAC)**. IMR-32 and Kelly cells were grown in arginine- and lysine-free RPMI with 10% dialyzed FBS supplemented with either [$^{13}C_6$, $^{15}N_2$] lysine (100 mg/l) or [$^{13}C_6$, $^{15}N_4$] arginine (100 mg/l) (Cambridge Isotope Laboratories, Inc.) (heavy population) or identical concentrations of isotopically normal lysine and arginine (light population) for at least six cell doublings. Heavy-labeled cells were incubated in THZ531 (400 nM) for 2 h and light-labeled cells were incubated in DMSO solvent as a control. After inhibitor treatment, cells were collected by trypsinization and counted. Equal numbers of heavy and light cells were mixed, washed twice in PBS, snap-frozen, and stored at −80 °C until lysis.

**Phosphopeptide purification**. Phosphopeptide enrichment was performed using titanium dioxide microspheres as previously described[55]. Briefly, lyophilized peptides were dissolved in 50% acetonitrile (ACN; Honeywell)/2 M lactic acid (Lee Biosolutions), incubated with 1.25 mg TiO2 microspheres (GL Sciences) per 1 mg peptide digest and vortexed at 75% power for 1 h. Microspheres were washed twice with 50% ACN/2 M lactic acid and twice with 50% ACN/0.1% TFA. Phospho-peptides were eluted with 50 mM $K_2HPO_4$ (Sigma) pH 10 (adjusted with ammonium hydroxide; Sigma). Formic acid (EMD) was added to the eluates to a concentration of 1.7%. The acidified phosphopeptides were desalted using a C18 solid-phase extraction (SPE) cartridge and the eluate was vacuum centrifuged to dryness.

**Offline HPLC pre-fractionation**. Approximately 120 μg phosphopeptides were resuspended in 0.1% TFA (Trifluoroacetic acid) and fractionated via penta-fluorophenyl chromatography as previously described[56]. The 48 collected fractions were reduced to 16 by combining every 16th fraction, vacuum centrifuged to dryness and stored at −80 °C prior to analysis by LC-MS/MS.

**LC-MS/MS analysis**. LC-MS/MS analysis was performed on an Orbitrap Fusion Tribrid mass spectrometer (ThermoFisher Scientific, San Jose, CA) equipped with an EASY-nLC 1000 ultra-high pressure liquid chromatograph (ThermoFisher Scientific, Waltham, MA). Phosphopeptides were dissolved in loading buffer (5% methanol (Fisher)/1.5 % formic acid) and injected directly onto an in-house pulled polymer coated fritless fused silica analytical resolving column (40 cm length, 100 μm inner diameter; PolyMicro) packed with ReproSil, C18 AQ 1.9 μm 120 Å pore (Dr. Maisch). Phosphopeptides in 3 μl loading buffer were loaded at 650 bar pressure by chasing onto the column with 10 μl loading buffer. Samples were separated with a 90-min. gradient of 4–33% LC-MS buffer B (LC-MS buffer A: 0.125% formic acid, 3% ACN; LC-MS buffer B: 0.125% formic acid, 95% ACN) at a flow rate of 330 nl/min. The Orbitrap Fusion was operated with an Orbitrap MS1 scan at 120 K resolution and an AGC target value of 500 K. The maximum injection time was 100 ms, the scan range was 350–1500 $m/z$ and the dynamic exclusion window was 15 s (±15 ppm from precursor ion $m/z$). Precursor ions were selected for MS2 using quadrupole isolation (0.7 $m/z$ isolation width) in a "top speed" (2 s duty cycle), data-dependent manner. MS2 scans were generated through higher energy collision-induced dissociation (HCD) fragmentation (29% HCD energy) and Orbitrap analysis at 15 K resolution. Ion charge states of +2 through +4 were selected for HCD MS2. The MS2 scan maximum injection time was 60 milliseconds and AGC target value was 60 K.

**Peptide spectral matching and bioinformatics**. Raw data were searched using COMET[57] against a target-decoy version of the human (*Homo sapiens*) proteome sequence database (UniProt; downloaded 2013; 20,241 total proteins) with a pre-cursor mass tolerance of ±1.00 Da and requiring fully tryptic peptides with up to 3 missed cleavages, carbamidomethyl cysteine as a fixed modification and oxidized methionine as a variable modification. For SILAC experiments, the additional masses of lysine and arginine isotope labels were searched as variable modifications. Phosphorylation of serine, threonine and tyrosine were searched with up to 3 variable modifications per peptide, and were localized using the phosphoRS algorithm[58]. The resulting peptide spectral matches were filtered to <1% false discovery rate (FDR) by defining thresholds of decoy hit frequencies at particular mass measurement accuracy (measured in parts per million from theoretical), XCorr and delta-XCorr (dCn) values.

**Antibodies**. The following antibodies were used: RNAPII CTD S2 (Bethyl cat# A300-654A, 1:10,000); RNAPII CTD S5 (Bethyl cat# A300-655A, 1:10,000); RNAPII (Santa Cruz cat# sc-899, 1:1000); RNAPII CTD S7 (Millipore cat# 041570, 1:10,000) RNAPII CTD Thr4 (Active Motif cat# 61361, 1:1000) cleaved PARP (Cell Signaling cat# 9541, 1:1000); β-Actin (Cell Signaling cat# 4967, 1:4000); CDK12 (Cell Signaling cat# 11973 s, 1:1000); CDK13 (Bethyl cat# A301-458A, 1:1000); CDK13 antibody (1:1000) used for WB in Supplementary Fig. 2e was kindly provided by Dr. Arno Greenleaf; γ-H2AX (Cell Signaling cat# 9718, 1:200), RAD51 (GeneText cat# GTX70230, 1:1000), BRCA1 (Cell signaling cat# 9010S, 1:500), BARD1 (Santa Cruz Biotechnology cat# sc11438, 1:500), Alexa-488 (Molecular Probes cat#A11008, 1:300).

**RT-PCR**. Total RNA was isolated with the RNAeasy Mini kit (QIAGEN). One μg of purified RNA was reverse transcribed using Superscript III First-Strand (Invitrogen) with random hexamer primers following the manufacturer's protocol. Quantitative PCR was carried out using the QuantiFast SYBR Green PCR kit (Qiagen) and analyzed on an Applied Biosystems StepOne Real-Time PCR System (Life Technologies). Each individual biological sample was qPCR-amplified in technical triplicate and normalized to GAPDH as an internal control. Relative quantification was calculated according to the ΔΔCt relative quantification method. Error bars indicate ± SD of three replicates. Primers sequences are available on request.

**RNA extraction and synthetic RNA spike-in for gene expression analysis**. Cells were treated with 400 nM of THZ531 or with DMSO for 6 h. Cell numbers were determined prior to lysis and RNA extraction. Biological duplicates (5 million cells per replicate) were collected and homogenized in 1 ml of TRIzol Reagent (Invitrogen) and purified with the mirVANA miRNA isolation kit (Ambion) following the manufacturer's instructions. Total RNA was treated with DNA-free™ DNase I (Ambion), spiked-in with ERCC RNA Spike-In Mix (Ambion), and analyzed on an Agilent 2100 Bioanalyzer (Agilent Technologies) for integrity. RNA was hybridized to Affymetrix GeneChip_ PrimeView Human Gene Expression arrays (Affymetrix).

**Transient transcriptome sequencing**. Cells were treated with DMSO or 400 nM of THZ531 for 30 min and 2 h. Cells were labeled in media for 10 min with 500 μM 4-thiouridine (4sU, Sigma-Aldrich). RNA extraction was performed with TRIzol (Ambion) following the manufacturers' instructions. Total RNA was treated with DNase I (Invitrogen). Subsequently, the purified RNA was fragmented on a BioRuptor Next Gen (Diagenode) at high power for one cycle of 30″/30″ ON/OFF. Fragmented samples were subjected to labeled RNA purification as previously described[59]. Labeled fragmented RNA was spiked-in with ERCC RNA Spike-In Mix (Ambion) and analyzed on an Agilent 2100 Bioanalyzer (Agilent Technologies) for integrity. Sequencing libraries were prepared with the RNA-seq library kit (TruSeq Stranded Total RNA RiboZero Gold, Illumina) as per the manufacturers' instructions. All samples were sequenced on a HiSeq 2500 sequencer.

**Poly(A) 3′-end sequencing**. Cells were treated with DMSO or THZ531 (400 nM in IMR-32 cells; 200 nM in Kelly and Kelly E9R cells) for 2 and 6 h. RNA extraction was performed with TRIzol (Ambion) following the manufacturers' instructions. Total RNA was treated with DNase I (Invitrogen). Sequencing libraries were prepared with the RNA-seq library kit (QuantSeq 3′ mRNA Sequencing REV, Lexogen) following the manufacturers' instructions. All samples were sequenced on a HiSeq 2500 sequencer.

**In vitro kinase assay**. Recombinant CDK12/CycK complex was prepared from baculovirus infected Sf9 cells as described[60]. Substrate proteins CstF64 (aa 509-577), CDC5L (aa 370-505), SPT6H (aa 1434-1544) and SF3B1 (aa113-462) were expressed as GST-fusion proteins in E. coli and purified to homogeneity. Radio-active kinase reactions were performed with 0.2 μM CDK12/CycK or CDK13/CycK and 100 μM each of substrate protein, and 1 mM ATP at 30 °C for 30 min in kinase buffer as described[60]. Reactions were spotted onto P81 Whatman paper squares, washed three times and radioactivity counted on a Beckman Scintillation Counter

(Beckman-Coulter) for 1 min. Measurements were performed in triplicate and are represented as mean ± SD.

**Peptide mass fingerprinting**. GST-tagged proteins were resolved on a 12% SDS-PAGE gel and stained with Coomassie brilliant blue. Protein bands were cut from the SDS-PAGE gel and submitted for mass spectrometry analysis to the Bioanalytical Mass Spectrometry Group, Max Planck Institute for Biophysical Chemistry, Gottingen, Germany.

**3′ RACE PCR**. Cells were treated with 400 nM of THZ531 or with DMSO for 6 h. RNA extraction was performed with TRIzol (Ambion) following the manufacturers' instructions. Total RNA was treated with DNase I (Invitrogen). Subsequently, total RNA was poly(A) selected using oligo-dT dynbeads (Invitrogen). RNA was reverse transcribed using 3′-RACE adapter oligonucleotide (FirstChoice RLM-RACE Kit; Life Technologies). Nested PCR was performed using Phusion High Fidelity DNA Polymerase (New England Biolabs). PCR products were resolved on a 1.3% agarose gel, purified using a gel extraction kit (Qiagen), and sequenced. PCR primer sequences available upon request.

**Gene expression analysis**. Microarray data were analyzed using a custom CDF file (GPL16043) that contained the mapping information of the ERCC probes used in the spike-in RNAs. The arrays were normalized according to previously described protocols[61]. Briefly, all chip data were imported in R (version 3.0.1) using the affy package[62], converted into expression values using the expresso command, normalized to take into account the different numbers of cells and spike-ins used in the different experiments and renormalized using loess regression fitted to the spike-in probes. Sets of differentially expressed genes were obtained using the limma package[63] and a false discovery rate (FDR) of 0.05. Statistical comparisons of distributions of fold changes were done using the Mann–Whitney U-test.

**TT-sequencing data processing**. For each sample, paired-end 75 bp reads were obtained and mapped to the human genome (GRCh38) and ERCC spike-in sequences. An average of 50 M (~90%) read pairs were mapped with STAR (version STAR_2.5.1b_modified) with default parameters. Only high quality and properly paired reads were retained for further analysis using samtools (v1.3.1) with parameters "-q 7 and –f 83,99,147,163". To normalize for library size and sequencing depth variation, individual spike-in reads were counted with *samtools idxstats*. This was used as input to calculate a sample-specific size factor with *estimateSizeFactorsForMatrix* (DESeq2). To create strand-specific sample coverage profiles in 100 bp bins, we used bamCoverage (DeepTools v2.5.4) with previously calculated size factors and parameters "--scaleFactor --normalizeUsingRPKM –filterRNAstrand –bs 100". Genome-wide correlation of biological replicates was calculated using Spearman's rank coefficient and visualized using scatterplots and heatmaps. These results showed high reproducibility for each condition and hence, for all analyses except differential expression and transcript usage, replicates were merged using *samtools merge* and processed again as described for the individual replicates.

**Poly(A) 3′-sequencing data processing**. For each sample single-end 100 bp reads were obtained and filtered using bbduk.sh from BBMap (v37.00) and parameters "k = 13 ktrim = r useshortkmers = t mink = 5 qtrim = r trimq = 20 minlength = 75 ref = truseq_rna.fa.gz" to remove potential adapter contamination or low-quality reads. High-quality reads were subsequently mapped to the human genome (GRCh38) using STAR (version STAR_2.5.1b_modified) and the following parameters "--outFilterType BySJout --outFilterMultimapNmax 20 --alignSJoverhangMin 8 --alignSJDBoverhangMin 1 --outFilterMismatchNmax 999 --outFilterMismatchNoverLmax 0.1 --alignIntronMin 20 --alignIntronMax 1000000 --alignMatesGapMax 1000000 --outSAMattributes NH HI NM MD --outSAMtype BAM SortedByCoordinate". To create strand-specific sample coverage profiles in 50 bp bins, we used bamCoverage (DeepTools v2.5.4) with parameters "--normalizeUsingRPKM –filterRNAstrand –bs 50". Genome-wide correlation of biological replicates was calculated using Spearman's rank coefficient and visualized using scatterplots and heatmaps. These results showed high reproducibility for each condition and hence, for all visualizations, replicates were merged using *samtools merge* and processed again as described for the individual replicates. Dataset-specific steps: (i) for the Kelly WT and Kelly E9R samples ERCC spike-in sequences were added and included in the downstream analysis to allow the detection of absolute gene expression level differences in another NB cell line. (ii) The CDK12/13 shRNA-mediated knockdown samples were processed in two different batches which displayed variable outcomes in sequence quality and depth, hence, to adjust for technical confounding effects, the samples were first randomly downsampled to the lowest individual library size.

**Poly(A) 3′-sequencing peak identification and filtering**. Poly(A)-seq peaks were called using MACS2 (v 2.1.1) and parameters "--nomodel --extsize 100 --shift 0" using a merged bam file of all samples. To identify false positive poly(A) peaks, two criteria were used: (1) the presence of the potential PAS motifs (AATAAA, ATTAAA, AGTAAA, TATAAA, AATATA, AATACA, CATAAA, GATAAA,

AATGAA, ACTAAA, AAGAAA, AATAGA) computed in a 100 bp window upstream of the peak in a strand-specific manner, and (2) the presence of a genomic 25-adenine (A) stretch with a maximum of 3 mismatches computed in a 50 bp window downstream of the peak in a strand-specific manner. Peaks were removed if they were not associated with a PAS motif but were associated with a genomic stretch of A's. Reads associated with these peaks were subsequently removed from the original mapped reads with the command "samtools –L regions_to_remove.bed –U output.bam". Strand specific coverage files were recomputed as described before. Retained Poly(A)-seq peaks were annotated in a step-wise manner; first, peaks were considered to be associated with the 3′ UTR if they were within the vicinity of the transcription end site (TES, −200 to +600 bp), next, the remaining peaks were considered to be intergenic or genic and, in the latter case, overlapping with an exon or intron. If a peak overlapped multiple transcripts, priority was given to protein-coding transcripts followed by longer transcripts. For metagene plots genes were represented by the isoform that showed the highest combined 3′-UTR expression level.

**Transcript selection and custom genome annotation**. Kallisto (v0.43.1) with parameters "--bootstrap-samples --rf-stranded" was used to determine the relative expression levels of all annotated transcripts as transcripts per million (TPM) (GencodeV27, GRCh38.p10). To reduce noise for downstream analyses, low-expressed genes (gene TPM < 2) or infrequently used transcripts (fraction of transcript < 0.2, except if transcript TPM > 5) were removed. A custom genome annotation was created by only retaining the detected transcripts. For each gene, all individual transcripts were merged using the reduce function of the GenomicRanges package in R to create a reduced exonic or intronic representation.

**Differential transcript usage**. Differential transcript usage between DMSO and THZ531-treated samples at 2 h was determined with the rats package in R using count estimates from Kallisto and further filtered based on our custom gene annotation.

**Alternative splicing**. To extract alternative splicing events the TT-seq paired-end reads were first re-mapped with STAR as described before, except soft-clipping was excluded by setting the parameter "--alignEndsType" to "EndToEnd" to favor reads spanning the exon-intron border. Next, we used the rMATS tool (v4.0.1) with the default settings to identify statistically significant alternative events (FDR < 0.05).

**Intron retention index**. The intron retention (IR) index is the log2 ratio of the exon and intron ratios calculated on the TT-seq normalized coverage in THZ531-treated and DMSO-treated samples. Only genes with a minimum of 10 exonic and 5 intronic TT-seq reads in either THZ531 or DMSO treated cells were included in this analysis. In short:

Exon ratio = (exon coverage THZ531 + 1)/(exon coverage DMSO + 1)
Intron ratio = (intron coverage THZ531 + 1)/(intron coverage DMSO + 1)
IR index = log2 (Intron ratio/Exon ratio)

The Fisher's exact test was used to determine significant intron loss (adjusted $p$-value < 0.05 and IR index < −1) or retention (adjusted $p$-value < 0.05 and IR index > 1).

**Sample-specific poly(A) 3′-seq peaks and genomic distribution**. To calculate sample-specific poly(A) 3′-seq peaks, only peaks with a $-\log_{10} q$-score ≥ 5 were retained. For each peak, overlapping reads were counted for each condition and a log-ratio score was calculated as log2 (THZ531_6h + 1/DMSO + 1). Peaks with a minimum number of 64 reads and a log-ratio score >1 or < −1 were considered THZ531_6h- and DMSO-specific respectively. Each peak was assigned to only one genomic region based on overlap with our custom gene annotation in a ranked order, i.e., annotated TES, exon, intron or intergenic.

**Differential expression**. Pairwise differential expression for exonic regions between DMSO-treated and THZ531-treated samples was calculated in the following manner: a 5′ upstream (−50 to −2050 bp of TSS) and 3′ downstream (+50 to +2050 bp of TES) 2 kb window was created and regions overlapping with genes on the same strand were removed. Only regions with a final minimum length of 200 bp were retained for further analysis. Genomic locations for exonic regions were converted to an saf format to calculate gene counts for each region using FeatureCounts (v1.5.0-p1). To detect differentially expressed genes for each genomic region the DESeq2 package in R was used with the size factors calculated previously (see TT-seq data processing). A gene with an absolute log2 fold-change > 1 and an adjusted $p$-value < 0.1 was considered significant. Genes with too few reads to perform differential analysis were considered not significantly changed.

**Differential global expression change of aggregated DDR gene set**. A combined set of genes that are part of the DNA damage response (DDR) was created by aggregating genes assigned to any DDR pathway in the databases found at https://www.mdanderson.org/documents/Labs/Wood-Laboratory/human-dna-repair-genes.html and http://repairtoire.genesilico.pl (see Supplementary Data 3). To

identify if these genes as a whole were more downregulated, 1000 random and equal-sized gene sets were generated and a distribution of the average level of expression change was plotted and compared to that of the initial DDR gene set to calculate a z-score.

**Gene biotype and size selection**. Gene biotypes assigned by Gencode were simplified in a two-step manner. First, only gene biotypes with a minimum of 20 members were considered. Next gene lengths of all detected non-coding genes (i.e., excluding protein-coding) were clustered using kmeans in two groups (short vs. long non-coding genes). Together, this resulted in three groups selected on biotype and gene length: (1) protein-coding genes (2) long non-coding genes (lincRNA, antisense_RNA, processed_transcript, sense_intronic, transcribed_unitary_pseudogene, TEC (to be experimentally confirmed), transcribed_processed_pseudogene, transcribed_unprocessed_pseudogene, unprocessed_pseudogene), and (3) short non-coding genes (snRNA, scaRNA, snoRNA, Mt_tRNA, misc_RNA, processed_pseudogene, rRNA). Protein-coding genes were further stratified into 4 length classes based on the quartiles of length distribution, i.e. long (>64.5 kb), medium-long (26.4–64.5 kb), medium-short (9.9–26.4 kb) and short (<9.9 kb). The latter group was further divided into three equal groups based on the 0.33, 0.66, and 1 quantiles of only the short gene length distribution, resulting in short (9.9–6.4 kb), very short (6.4–3.4 kb), and ultra-short (<3.4 kb) groups. Simple linear regression and Spearman correlation coefficient between log2 scaled length and log2 fold-change of exonic reads for genes was performed in R.

**Metagene profiles**. A gene metaprofile was created by dividing each gene (from TSS to TES) into 50 equally sized bins; 2 kb upstream and downstream flanking regions were binned in bins of 100 bp. Bedgraph files with normalized reads from TT-seq or poly(A) 3′-seq were used to calculate read density (RPM/bp) across those bins and subsequently summarized for all genes. To create a TSS or TES metaprofile, we followed an analogous approach with variable upstream and downstream flanking regions and summarized bins of 50 bp. To compare TT-seq and poly(A) 3′-seq profiles, calculated read densities were rescaled between 1 and 100.

**Inference of proximal RNA polymerase dynamics**. Transcription dynamics were calculated in 50 bp bins in the 500 bp upstream and 10 kb downstream flanking regions of the TSS. Change in read accumulation was calculated by normalized TT-seq read subtraction [THZ531 – DMSO] and the first derivative was computed to obtain the rate of accumulation change. To overlay both calculations, the rate of change was rescaled to the minimum and maximum of the change values and a loess smoothing curve was then fitted for visualization purposes.

**Correlation of transcript length and 3′ expression changes**. To identify differential expressed genes based on 3′ poly(A)-sequencing, all counts for 3′ UTR-associated polyadenylation sites were summarized per gene. This data matrix was log2 normalized and used to identify differential expression and fold-changes with the limma package in R. Correlation between fold-changes and transcript length was performed on the highest expressed transcript for each gene in the control condition. A generalized additive model (GAM) smoothing curve was fitted to each treatment to observe global changes and for visualization purposes.

**PCPA analysis**. Treatment-induced PCPA for protein-coding genes was calculated as in Oh et al.[37] with minor modifications. To determine whether a gene exhibits a coverage profile expected with PCPA, two scores were calculated. First, for each gene, we calculated an exon-score to determine if there was an increased loss of reads at the last exon compared to the first exon with THZ531 treatment: last exon [log2 (THZ531_2h/DMSO)] – first exon [log2 (THZ531_2h/DMSO)]. Next, an iQ-score to determine if there was an increased number of reads in the first quarter (iQ1) of the region between a gene's first 5′ splice site and the last 3′ splice site and the last quarter (iQ4) was calculated for each gene in the THZ531-treated samples: log2(iQ1/iQ4). Genes were considered to undergo THZ531-induced PCPA with an exon-score < −1 and an iQ-score >1.

**Intronic polyadenylation usage**. For each transcript (TPM > 1) the reads of all intronic and 3′ UTR-associated poly(A) sites were summarized. To compare the change and usage of intronic versus 3′ UTR-associated poly(A) sites between different treatments, an odds ratio (OR) was calculated for each treatment sample but excluding transcripts that had no intronic poly(A) sites in either treatment. A two-sample Kolmogorov-Smirnov test was then used to detect changes in OR distributions between different treatments.

**Correlation of transcript length and number of intronic polyadenylation sites**. To identify the relationship between the number of identified polyadenylation sites and transcript length, a polynomial regression curve (y ~ poly(x,2)) was fitted for all genes or DDR genes only. A Wilcoxon Rank Sum test was used to determine if the difference between predicted values for DDR genes between the two models [prediction DDR – prediction all genes] was significantly different.

**Genetic determinants analysis**. Known U1 (GGTGAG, GGTAAG and GTGAGT), PAS (AATAAA) motifs and GC content percentages were computed for each gene along the entire gene axis (TSS to TES) using the Biostrings package in R. A Wilcoxon Rank Sum test and Cohen's $d$ effect size were used to determine individual differences between all genes and genes with PCPA for each selected genetic determinant (gene length, length of first intron, number of introns, GC content, expression and ratio between U1 and PAS).

**Splice site conservation analysis**. Calculation of splice site conservation scores was performed as previously described[34] with modifications. In brief, position weight matrices (PWMs) for 5′ and 3′ splice sites were created using all introns that contain the established 5′ GT and 3′ AG sequence. For the 5′ and 3′ splice sites (ss) 9 (−2 bp:6 bp of 5′ss) and 15 bp (−14 bp of 3′ss) respectively were used. Next, introns with and without intronic polyadenylation sites (intronic poly(A) > 0) were scored for both the 5′ and 3′ PWM for their respective splice sites. A combined score for each intron was computed by summarizing the scores of the 5′ and 3′ splice site. The Wilcoxon Rank Sum test and Cohen's $d$ effect size were used to determine biologically meaningful differences between introns with and without an intronic polyadenylation.

**Enrichment analysis**. Gene ontology enrichment for selected gene sets was performed using the enrichR package in R. The Enrichr score[64] is the combined score of the adjusted $p$-value and the $z$-score using the Fisher's exact test. Enrichment of individual gene sets was considered significant if the adjusted $p$-value < 0.01, unless stated otherwise. The Fisher's exact test was used to determine significant overlap between other publicly available datasets.

**Genomic visualization**. To visualize coverage tracks a custom build visualization tool was used (github.com/RubD/GeTrackViz2).

**Reporting summary**. Further information on experimental design is available in the Nature Research Reporting Summary linked to this article.

## Data availability
Microarray, TT-seq, Poly(A) 3′-seq datasets have been deposited in the Gene Expression Omnibus (GEO), accession number GSE113314. The SILAC dataset has been deposited in the ProteomeXchange Consortium, accession number PXD009533. All other data are available from the corresponding author upon request. Data underlying Figs. 1, 6 and Supplementary Figs. 1–4 are provided as a Source Data file.

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

## Acknowledgements

We thank K. Adelman, T. Henriques, P. Cramer, S. Gressel and A. Meyer for detailed protocols, advice on experimental design and helpful discussions. We thank the Whitehead Institute Genome Core for RNA sequencing. We thank members of our laboratory for helpful discussions. This work was supported by NIH R01CA197336 (to R.E.G. and R.A.Y.), Friends for Life Neuroblastoma Foundation (to R.E.G.), Alex's Lemonade Stand Foundation (to R.E.G. and O.B.D.), the DFCI Claudia Adams Barr Award (to R.E.G. and N.S.G.) and the Rally Foundation for Childhood Cancer Research (to M. K.). The Gray laboratory is supported by the DFCI Hale Family Pancreatic Center and NIH grants, CA154303-06 and CA179483-02.

## Author contributions

M.K. designed and carried out molecular, cellular and genomic experiments. R.D. performed bioinformatics analyses. A.V.G. performed the SILAC experiments under the supervision of S.A.G. S.D. performed the in vitro kinase assays under the supervision of M.G. D.S.D. analyzed the microarray expression data. Y.G. established Kelly E9R cells and performed target engagement experiments with CDK13. N.K. provided protocols and assistance with target engagement studies. M.P. provided technical support with generation of CDK12/13 shRNA knockdowns. B.S. performed 3′ RACE PCR and provided technical support. E.C. performed the initial cell viability assays of THZ531. H.H. generated the consensus sequences from the SILAC data. O.B.D. performed viability assays under the supervision of M.K. T.H.Z. generated THZ531. A.L.G. provided CDK13 antibody and advice. G.-C.Y. supervised the bioinformatics analyses. N.S.G. and R.A.Y. provided feedback on study design and experimental results. M.K., R.D., and R.E.G. wrote the manuscript. R.E.G. conceived the project and supervised the research. All authors discussed the results and commented on the manuscript.

## Additional information

**Competing interests:** N.S.G. is a founder and equity holder of Syros Pharmaceuticals, C4 Therapeutics, Petra Pharma, Gatekeeper Pharmaceuticals and Soltego. R.A.Y. is a founder and shareholder of Syros Pharmaceuticals, Camp4 Therapeutics, Omega Therapeutics and Dewpoint Therapeutics. R.E.G. is on the SAB of Global Gene Corp. The remaining authors declare no competing interests.

