## [Peer Review File · Nature Communications]

REVIEWERS' COMMENTS:

Reviewer #1 (Remarks to the Author):

I thank the authors for their response to the previous comments. I think they've done an excellent job in revising the manuscript. It is more clear and the conclusions are now better aligned with the data. I have some remaining concerns that I think are important and should be addressed.

- In the first heading of the results section, "and affects distal transcriptional elongation" should be removed, as it is not addressed in this section. The changes in CTD phosphorylation cannot be considered sufficient to demonstrate this point.
- "CDK13" should be added to the heading on page 9: "CDK12 inhibition leads to premature cleavage and polyadenylation". Otherwise this heading is a bit misleading.
- Similarly, on page 10, the heading should read: "The termination defect seen with THZ531 is primarily due to inhibition of CDK12"
- In Figure 2b, more information is needed about the "rate of change" analysis. Some detail is provided in the methods, but it is important to demonstrate the reliability of this method. I have not seen such an analysis previously. To help convince readers that the method is robust, the authors should test on other published datasets to determine if the method yields predictable and reliable results. How do the data look upon comparison of the DMSO TT-seq replicates?
- In Figure 6d, IC50 values should be shown on the plots.
- The data in Figure 6e should be removed. Too premature and not convincing. Seems like a tangential direction relative to the rest of the manuscript. The authors describe the data as showing "enlarged nuclear speckles" in THZ531 cells, but comparing with the DMSO image shown, the speckles appear much larger and diffuse with DMSO. Related data are shown in SFig 8f and this should be removed as well.
- A new representative spectrum should be shown to replace SFig 8e. The current example is for SPT6, but that was shown to be an artifact (phosphorylation occurs in the absence of added kinase, SFig 8d). Also, the spectrum is not labeled in a way that clarifies the phospho-site for the readers.
- In the discussion, on page 16, a paragraph starts with "Inhibition of CDK12 by THZ531 results in increased transcription initiation..." This statement needs to be modified. Increased transcription initiation has not been convincingly demonstrated here. Later in the same sentence, "abortive release" is mentioned and the reference cited (55) deals with termination after promoter escape, at the promoter proximal pause region. The phrase "abortive release" is commonly used to refer to abortive initiation prior to promoter escape. To avoid confusion, I suggest the authors modify this wording.
- In this same paragraph (page 16 and into page 17), reference 55 is cited throughout and I think this is counterproductive. Reference 55 is a fine paper, but the focus of that study is promoter-proximal pausing. The authors' characterization of the "peak observed at ~1000nt downstream" is likely something entirely different and this distinction should be made clear to the reader.
- Also in the same paragraph, it is stated that "We also observed a sharp decrease in the accumulation of TT-seq reads at the start of elongation" The authors need to define what is meant by "the start of elongation" as many will consider this to be past the pause site at around +100. Later in

the paragraph it is stated that "Thus, we postulate that in the absence of CDK12 activity, Pol II is unable to enter into productive elongation;" It is more accurate to include CDK13 with CDK12 here and also more accurate to state that RNAPII is "less capable" of entering productive elongation, based upon the experiments completed and the data shown.

- In the middle paragraph on page 17, it is stated that "CDK12 results not only in a defect in elongation, but also, increased transcription initiation...". As noted above, an increase in initiation was not directly measured but can only be inferred from the data. This statement should be re-phrased. Also, in the next sentence it is stated that CDK9 and CDK12 have "opposite" effects. A better word would be "different" or "non-redundant" effects?
- I suggest that the paragraph on page 17, beginning with "The RNA processing defects" be removed, and the first 2 sentences of the following paragraph (beginning with "One plausible explanation") should be removed. I don't think this text adds much to the discussion since RNAPII rates were not measured. The discussion is speculative but it does not read that way. An alternative is to shorten this section and state that changes in RNAPII elongation rates may occur upon CDK12/13 inhibition and if true (remains to be tested) then this could potentially contribute to the intronic polyA measured here.
- The authors are likely aware of a related study published by the Sharp lab (Dubbury et al. Nature 2018). This should be cited and the results compared in the discussion.

Reviewer #2 (Remarks to the Author):

The authors have addressed all my concerns. I have no additional comments.

Manuscript: NCOMMS-18-16326-T CDK12 loss in cancer cells affects DNA damage response genes through premature cleavage and polyadenylation

We thank the reviewers for the effort and time put into the review of the manuscript. We especially appreciate reviewer 1's constructive comments which have all been addressed below and have significantly improved the manuscript. A point by point response to each comment is given below with changes to the manuscript indicated here and underlined in the manuscript.

Reviewer #1

1. In the first heading of the results section, “and affects distal transcriptional elongation” should be removed, as it is not addressed in this section. The changes in CTD phosphorylation cannot be considered sufficient to demonstrate this point.

Response: We agree with the reviewer's comment and have removed this phrase.

2. “CDK13” should be added to the heading on page 9: “CDK12 inhibition leads to premature cleavage and polyadenylation”. Otherwise this heading is a bit misleading.

Response: We have added “CDK13” to the heading on page 9.

3. Similarly, on page 10, the heading should read: “The termination defect seen with THZ531 is primarily due to inhibition of CDK12”

Response: We have amended the heading on page 10 as suggested.

4. In Figure 2b, more information is needed about the “rate of change” analysis. Some detail is provided in the methods, but it is important to demonstrate the reliability of this method. I have not seen such an analysis previously. To help convince readers that the method is robust, the authors should test on other published datasets to determine if the method yields predictable and reliable results. How do the data look upon comparison of the DMSO TT-seq replicates?

Response: We provide a detailed explanation of the analysis in Figure 2b based on the following: (1) clarification of the procedure, (2) a comparison with another common procedure and its potential pitfalls, (3) applicability to another dataset and (4) statistical robustness with respect to biological replicates. In the response below, a **bold figure callout** refers to figures in the response letter, while an underlined figure callout indicates those in the manuscript. A summarized version of the analysis method has also been added to the Methods section and both Figure 2b and Supplemental Figure 4a have been replaced by slightly modified versions that are less prone to interpretation errors (see next paragraph).

1) Procedure steps and details

The plots in Figure 2b and Supplemental Figure 4 in the manuscript are extensions of a common gene metaplot (as shown in **Figure 1**) that shows the coverage of nascent transcripts generated within the 10 min window during which the cells were labeled with 4sU (see TT-seq method).

The extension consists of calculating the difference between THZ531 and DMSO treated cells (**Figure 2a**, red curve) and subsequently calculating the first derivative of this difference (**Figure 2a**, blue curve) to determine whether differences in transcript accumulation are increasing or decreasing (i.e. rate of change). In the original version of the manuscript (Figure 2b and Supplemental Figure 4) both the difference and rate of change were rescaled between 0 and 1 to visualize both dynamics within the same range and in the same plot; however, we have now modified this figure by rescaling the rate of change to the minimum and maximum of the difference (**Figure 2b**). This approach results in identical conclusions to that of our original figure but is less prone to interpretation errors because positive and negative y-values of the difference can now be directly interpreted as accumulation or loss of read density, respectively, in THZ531-treated samples. We have corrected one such interpretation error for Supplementary Figure 4, where we had previously stated that net average loss of read

density is around 10 kb 3' of the TSS, while this is in fact around 6 kb. For visualization purposes we also fitted a loess curve to the individual values, however this is not absolutely required.

As such, this procedure helps to understand the dynamics of transcript accumulation between two conditions without having to fine tune any parameters or make any assumptions. Hence, this procedure is, by definition, unbiased and reliable in its outcome, although its usefulness is dependent on the quality of the data. The dependence on high quality data is noticeable in the dataset that we used to demonstrate the functionality of this procedure¹, but can be alleviated by using loess-predicted values and modifying the calculations of the first derivate (see #3 for further details).

2) Comparison with polymerase II pausing index calculations

The rationale for using this type of analysis and visualization originated not only from the fact that it was parameter- and assumption-free as described in the previous paragraph, but also because we believe that other methods such as calculating the polymerase II (Pol II) pausing index (PPI) are more prone to certain pitfalls that we have faced during the analysis of the data presented in this manuscript:

Figure 1. Metaplot depicting the TT-seq coverage profiles of DMSO and THZ531 treated cells.

Figure 2. Metaplots depicting the TT-seq coverage difference (red) and rate of change (blue) between THZ531 and DMSO treated cells. Original (a) and modified (b) plot.

a) The PPI is calculated as the ratio of accumulated reads around the TSS and the gene body. Changes in PPI between 2 experimental conditions (either a ratio or difference) can then provide insights into how Pol II dynamics changes genome-wide. However, there are no uniform standards of defining the TSS or gene body region and as such, there exists a wide range of values that have been used in recent years.

b) The PPI is based on the assumption that there are two discrete genomic regions that can be compared with each other; by contrast, our procedure shows the continuous changes and dynamics that occur during the initial phases of Pol II initiation, pausing and elongation.

c) In cases where there is an elongation defect, the PPI method will not be able to directly discriminate between the effects around the TSS or the gene body, which could result in inaccurate conclusions.

3) Applicability to another dataset

To demonstrate the applicability of our procedure to another dataset, we selected the recent paper from Fitz et al¹, where the authors show that Spt5, a conserved Pol II-associated pausing and elongation factor, has a primary effect on elongation in a specific genomic window (15-20 kb) which is related to the acceleration of Pol II during the initial phase of elongation. Hence, we expected to see a similar or delayed dynamic profile using our method for analysis of their data.

change based on the loess-fitted predicted data of the coverage differences (**Figure 3c**, green curve). Based on small (**Figure 3c**, -0.5kb to +1kb) and large (**Figure 3d**, -1kb to +10kb) genomic windows around the TSS, we arrive at similar conclusions as Fitz et al¹. First, we observe that there is a marked increase in transcript accumulation at the beginning of elongation, however in contrast to our data, this accumulation does not immediately result in a significant drop after reaching a peak (around 1-1.5kb) but continues beyond the +10kb mark. Therefore, we believe that our procedure to visualize dynamic patterns around the TSS is robust and generates predictable results.

4) Statistical robustness of biological replicates

To evaluate whether these results were reproducible, we processed the 3 replicates separately and recreated a gene metaplot showing the coverage profile for each replicate (**Figure 4a**) and the accompanying coverage accumulation plot (**Figure 4b**), including a confidence interval based on the standard deviations. Together these analyses illustrate that the observations are robust and reproducible among biological replicates.

First, we downloaded the processed bigwig files from Fitz et al¹ and selected all TSS from genes with detectable expression (transcripts per million > 10). Although this is most likely not identical to the analysis performed by Fitz et al, it should not affect the global results if the data are robust enough. Next, we applied our procedure and noted that the accumulation of nascent reads between Spt5 KO and WT cells (**Figure 3a**, red curve) showed a similar trend to that seen in our data (**Figure 2b**, manuscript). However, the rate of change (**Figure 3a**, blue curve) displays a very irregular and dispersed pattern, which is most likely due to the very low coverage of the data and differences between replicates, as can clearly be seen for a representative individual gene (**Figure 3b**, also selected in the Fitz et al paper). These issues can be alleviated by calculating the rate of

5. In Figure 6d, IC₅₀ values should be shown on the plots.

Response: IC₅₀ values are now shown on the plots in the revised **Figure 6d**.

6. The data in Figure 6e should be removed. Too premature and not convincing. Seems like a tangential direction relative to the rest of the manuscript. The authors describe the data as showing “enlarged nuclear speckles” in THZ531 cells, but comparing with the DMSO image shown, the speckles appear much larger and diffuse with DMSO. Related data are shown in SFig 8f and this should be removed as well.

Response: We agree with the reviewer that the data presented in Figure 6e are too premature, therefore as suggested we have removed these figures.

7. A new representative spectrum should be shown to replace SFig 8e. The current example is for SPT6, but that was shown to be an artifact (phosphorylation occurs in the absence of added kinase, SFig 8d). Also, the spectrum is not labeled in a way that clarifies the phospho-site for the readers.

Response: We apologize that the labelling was not clear and have replaced this figure panel with a new representative spectrum for CDC5L that is labeled more clearly. We have also removed SPT6H as it could not be confirmed in subsequent studies.

8. In the discussion, on page 16, a paragraph starts with “Inhibition of CDK12 by THZ531 results in increased transcription initiation...” This statement needs to be modified. Increased transcription initiation has not been convincingly demonstrated here. Later in the same sentence, “abortive release” is mentioned and the reference cited (55) deals with termination after promoter escape, at the promoter proximal pause region. The phrase “abortive release” is commonly used to refer to abortive initiation prior to promoter escape. To avoid confusion, I suggest the authors modify this wording.

Response: We thank the reviewer for drawing our attention to these discrepancies. We have amended the above sentence in the manuscript as follows: “The CDK12-mediated transcriptional effects reported here agree with - but differ mechanistically from – those reported with inhibition of the other Pol II Ser2 elongation kinase, CDK9, where increased Pol II pausing leads to a defect in elongation and negatively impacts transcription initiation^{2,3}. By contrast, perturbation of CDK12, while also resulting in an elongation defect, is likely to be associated with increased transcription initiation and Pol II pause release. This conclusion is supported by the accumulation of nascent RNA reads both upstream (antisense) and at the TSS, that extended well beyond the Pol II pause sites to average peak densities ~+1000 bp downstream of the TSS, and especially by the continued rate of increase in signal accumulation beyond the pause sites. Thus, given that the accumulation of TT-seq reads coincided with the onset of productive elongation during which Pol II accelerates dramatically^{4,5}, it is likely that CDK12/13 function is critical for the recruitment and/or modification of components of the transcription machinery that together, sustain efficient rates of productive elongation⁶. Alternatively, or concomitantly, the accumulation of nascent RNA reads downstream of the TSS could indicate the existence of CDK12/13-dependent elongation checkpoints similar to those reported for CDK9⁷. We also observed a sharp decrease in TT-seq reads +1000 bp downstream of the TSS, with resultant early termination, a finding that was supported by increased poly(A) 3'-seq reads at the 5' proximal ends of these genes. Thus, we postulate that in the absence of CDK12/13 activity, Pol II is less capable of entering into productive elongation; instead, as previously proposed⁸, it is gradually released from chromatin, likely increasing the pool of free Pol II molecules that can engage in transcription initiation, and accounting for the increased TT-seq reads at the TSS in THZ531-treated cells.”

9. In this same paragraph (page 16 and into page 17), reference 55 is cited throughout and I think this is counterproductive. Reference 55 is a fine paper, but the focus of that study is promoter-proximal pausing. The authors' characterization of the “peak observed at ~1000nt downstream” is likely something entirely different and this distinction should be made clear to the reader.

Response: We agree with the reviewer that the observed peak accumulation 1000 bp from the TSS is not related to promoter-proximal pausing. We speculate that these signals may overlap with the acceleration phase of Pol II after pause release, which we term “productive elongation”⁶ or alternatively indicate the existence of a more specific CDK12-dependent elongation checkpoint as has been previously reported for CDK9 by Laitem et al.⁷.

We assume that the 2 mechanisms are not mutually exclusive and that the position of this checkpoint will depend on the interplay between Pol II and pre-mRNA processing factors.

10. Also in the same paragraph, it is stated that “We also observed a sharp decrease in the accumulation of TT-seq reads at the start of elongation” The authors need to define what is meant by “the start of elongation” as many will consider this to be past the pause site at around +100. Later in the paragraph it is stated that “Thus, we postulate that in the absence of CDK12 activity, Pol II is unable to enter into productive elongation;” It is more accurate to include CDK13 with CDK12 here and also more accurate to state that RNAPII is “less capable” of entering productive elongation, based upon the experiments completed and the data shown.

Response: We apologize for the ambiguity of our statement and have rephrased it as follows: “We also observed a sharp decrease in TT-seq reads +1000 bp downstream of the TSS, with resultant early termination, a finding that was supported by increased poly(A) 3'-seq reads at the 5' proximal ends of these genes.”. We have also clarified our terminology throughout the manuscript with regards to the phase of elongation based on the review by John Lis⁶, we use “productive elongation”, to mean transcription by Pol II past the pause site and “early elongation” to refer to the stage between initiation and productive elongation.”

We have also included CDK13 with CDK12 and have revised the sentence as per the reviewer’s suggestion, “Thus, we postulate that in the absence of CDK12/13 activity, Pol II is less capable of entering into productive elongation; instead, as previously proposed⁸, it is gradually released from chromatin, likely increasing the pool of free Pol II molecules that can engage in transcription initiation, and accounting for the increased TT-seq reads at the TSS in THZ531-treated cells.”

11. In the middle paragraph on page 17, it is stated that “CDK12 results not only in a defect in elongation, but also, increased transcription initiation...”. As noted above, an increase in initiation was not directly measured but can only be inferred from the data. This statement should be re-phrased. Also, in the next sentence it is stated that CDK9 and CDK12 have “opposite” effects. A better word would be “different” or “non-redundant” effects?

Response: We agree with the reviewer that we have not provide sufficient experimental evidence for increased initiation following CDK12 inhibition as well as the wording regarding the effects of CDK9 and CDK12. This section has been rewritten and has been replaced with the paragraph shown in the response to comment #8.

12. I suggest that the paragraph on page 17, beginning with “The RNA processing defects” be removed, and the first 2 sentences of the following paragraph (beginning with “One plausible explanation”) should be removed. I don’t think this text adds much to the discussion since RNAPII rates were not measured. The discussion is speculative but it does not read that way. An alternative is to shorten this section and state that changes in RNAPII elongation rates may occur upon CDK12/13 inhibition and if true (remains to be tested) then this could potentially contribute to the intronic polyA measured here.

Response: We agree and therefore, as suggested, have removed this paragraph and the first 2 sentences of the following paragraph from the manuscript.

13. The authors are likely aware of a related study published by the Sharp lab (Dubbury et al. Nature 2018). This should be cited and the results compared in the discussion.

Response: We are aware of this paper and compare our results with those of the above-mentioned paper as follows, “Dubbury et al⁹, recently examined the later effects of CDK12 depletion on total RNA expression in mouse ES cells, showing that CDK12 suppresses intronic polyadenylation as a mode of DDR gene regulation. The authors found that this mechanism is preserved in ovarian and prostate tumors with CDK12 loss-of-function mutations or deletions. Our findings augment those of the Dubbury study by (i) linking the unique susceptibility of DDR genes to CDK12 inhibition with their relatively longer lengths, lower GC content and lower ratios of U1 snRNP binding sites to intronic polyadenylation sites, and (ii) showing that these transcriptional effects occur at the nascent RNA level as early as 2 hours after CDK12 loss. Thus, effects attributed to CDK12 loss do not appear to be restricted to cancers with loss-of-function mutations, but to encompass those with severe underlying DNA damage, such as NB.”

References

- 1 Fitz, J., Neumann, T. & Pavri, R. Regulation of RNA polymerase II processivity by Spt5 is restricted to a narrow window during elongation. *EMBO J* **37**, doi:10.15252/embj.201797965 (2018).
- 2 Shao, W. & Zeitlinger, J. Paused RNA polymerase II inhibits new transcriptional initiation. *Nat Genet* **49**, 1045-1051, doi:10.1038/ng.3867 (2017).
- 3 Gressel, S. *et al.* CDK9-dependent RNA polymerase II pausing controls transcription initiation. *Elife* **6**, doi:10.7554/eLife.29736 (2017).
- 4 Jonkers, I., Kwak, H. & Lis, J. T. Genome-wide dynamics of Pol II elongation and its interplay with promoter proximal pausing, chromatin, and exons. *Elife* **3**, e02407, doi:10.7554/eLife.02407 (2014).
- 5 Danko, C. G. *et al.* Signaling pathways differentially affect RNA polymerase II initiation, pausing, and elongation rate in cells. *Mol Cell* **50**, 212-222, doi:10.1016/j.molcel.2013.02.015 (2013).
- 6 Jonkers, I. & Lis, J. T. Getting up to speed with transcription elongation by RNA polymerase II. *Nat Rev Mol Cell Biol* **16**, 167-177, doi:10.1038/nrm3953 (2015).
- 7 Laitem, C. *et al.* CDK9 inhibitors define elongation checkpoints at both ends of RNA polymerase II-transcribed genes. *Nat Struct Mol Biol* **22**, 396-403, doi:10.1038/nsmb.3000 (2015).
- 8 Steurer, B. *et al.* Live-cell analysis of endogenous GFP-RPB1 uncovers rapid turnover of initiating and promoter-paused RNA Polymerase II. *Proc Natl Acad Sci U S A* **115**, E4368-E4376, doi:10.1073/pnas.1717920115 (2018).
- 9 Dubbury, S. J., Boutz, P. L. & Sharp, P. A. CDK12 regulates DNA repair genes by suppressing intronic polyadenylation. *Nature* **564**, 141-145, doi:10.1038/s41586-018-0758-y (2018).